# CircFNDC3B regulates osteoarthritis and oxidative stress by targeting miR-525-5p/HO-1 axis

Zizheng Chen[1,2,3,4], Yizhen Huang[1,2,3,4], Yu Chen[1,3,4], Xiaodong Yang[1,3], Jinjin Zhu[1,2,3], Guang Xu[1,3], Shuying Shen[1,2], Ziang Hu[1,2], Peihua Shi [1,2✉], Yan Ma [1,2✉] & Shunwu Fan [1,2✉]

Osteoarthritis (OA) is a common chronic degenerative joint disease associated with a variety of risk factors including aging, genetics, obesity, and mechanical disturbance. This study aimed to elucidate the function of a newly discovered circular RNA (circRNA), circFNDC3B, in OA progression and its relationship with the NF-κB signaling pathway and oxidative stress. The circFNDC3B/miR-525-5p/HO-1 axis and its relationship with the NF-κB signaling pathway and oxidative stress were investigated and validated using fluorescence in situ hybridization, real-time PCR, western blotting, immunofluorescence analysis, luciferase reporter assays, pull-down assays, and reactive oxygen species analyses. The functions of circFNDC3B in OA was investigated in vitro and in vivo. These evaluations demonstrated that circFNDC3B promotes chondrocyte proliferation and protects the extracellular matrix (ECM) from degradation. We also revealed that circFNDC3B defends against oxidative stress in OA by regulating the circFNDC3B/miR-525-5p/HO-1 axis and the NF-κB signaling pathway. Further, we found that overexpression of circFNDC3B alleviated OA in a rabbit model. In summary, we identified a new circFNDC3B/miR-525-5p/HO-1 signaling pathway that may act to relieve OA by alleviating oxidative stress and regulating the NF-κB pathway, resulting in the protection of the ECM in human chondrocytes, highlighting it as a potential therapeutic target for the treatment of OA.

[1] Department of Orthopaedic Surgery, Sir Run Run Shaw Hospital, Zhejiang University School of Medicine, 3 East Qingchun Road, Hangzhou 310016 Zhejiang Province, China. [2] Key Laboratory of Musculoskeletal System Degeneration and Regeneration Translational Research of Zhejiang Province, Hangzhou 310016 Zhejiang Province, China. [3] Zhejiang University School of Medicine, Hangzhou 310016, China. [4]These authors contributed equally: Zizheng Chen, Yizhen Huang, Yu Chen. ✉email: peihua_shi@zju.edu.cn; zjumayan@zju.edu.cn; 0099203@zju.edu.cn

Osteoarthritis (OA) is one kind of common chronic joint disease, which is featured by the degeneration of articular cartilage, subchondral bone sclerosis, osteophyte formation and synovial hyperplasia[1–4]. A most recent study in 2019 reveals that 250 million people are suffering from OA worldwide[5]. Joint pain is the most common clinical symptom, and the knee is the most common site for OA, followed by the hand and hip. OA reduces the quality of life of patients and creates a heavy socioeconomic burden for the sufferers and their families. Identified risk factors include gender, age, previous joint injury, adiposity, heavy physical activity, and genetics amongst others[6–9]. Inflammatory, mechanical and metabolic factors are all involved in OA[5]. However, its pathogenesis is extremely complex and many of the nuances remain unknown. Consequently, there are few effective therapies available for OA, meaning that deeper insights into the pathogenesis of OA are urgently needed if we want to provide new therapeutic strategies for this condition.

Oxidative stress has been identified as a significant factor in the progression of various diseases, including OA[10]. A growing number of studies have shown that antioxidants can reduce OA severity, but there is still much to be discovered about their chondroprotective mechanisms in joint tissues. Increased reactive oxygen species (ROS) and decreased antioxidants result in increased oxidative stress and thus the activation of various catabolic factors, including extracellular matrix (ECM)-degrading proteases[11]. A recent study showed that antioxidant enzymes such as heme oxygenase-1 (HO-1) can help to resist the ROS-mediated damage[12] associated with OA and several other types of arthritis[10,13–15]. A recent article on *Aging* described a decrease in phospho-P65 (p-P65) levels in HO-1-overexpressing cells, suggesting that this protein may act to inhibit NF-κB-mediated effects under certain conditions[16]. Thus, HO-1 may be a potential therapeutic target for OA.

Several studies have linked the altered expression of micro-RNAs (miRNAs) to multiple disease processes, including OA[17]. miRNAs are short endogenous non-coding RNAs of 19–25 nucleotides in length which can function as posttranscriptional gene expression regulators[18]. A recent study showed that miR-1271 was upregulated in OA tissues and that miR-1271 suppresses the expression of ERG (E26 transformation-specific-related-gene), which is associated with OA progression[19]. Meanwhile, RNA sequencing and bioinformatics technologies have identified an abundance of circular RNAs (circRNAs). CircRNAs are a class of noncoding RNAs and their most important feature is a closed-loop structure that links the 3′ and 5′ ends allowing them to avoid exonucleolytic degradation by RNase R[20]. Current studies have identified that circRNAs often encode multiple miRNA-binding sites and act as miRNA sponges or endogenous RNAs (ceRNAs). For example, CircSERPINE2 targets miR-1271 and ETS-related gene[19], CircCDK14 could sponges miR-125a-5p and promotes Smad2 expression[21] and both protect against OA. These studies suggest a potential role of circRNAs in various biological processes.

At present, relatively few studies have focused on the relationship between circRNAs, oxidative stress, and OA. In this study, we identified hsa_circ_0001361 (also called circFNDC3B) and evaluated its expression in OA and revealed its mechanism of regulation in oxidative stress and OA progression via the circFNDC3B/miR-525-5p/HO-1 axis and its activation of the NF-κB signaling pathway. We believe that our elucidating the CircFNDC3B/miR-525-5p/HO-1 axis poses opportunities for the integration of multiple targets for the molecular therapy of OA.

## Results

### Characterization and expression analysis of CircFNDC3B in human OA and control tissues

We produced a circRNA profiling database from three clinical human OA and three control tissues in a previous study[19] where we identified a total of 12738 circRNAs. Both the OA and control groups demonstrated differential expression patterns for various circRNAs (Fig. 1a) and we focused on circRNA hsa_circ_0001361 (also called circFNDC3B) in this study. CircFNDC3B was transcribed from the human FNDC3B gene locus and was significantly downregulated in human OA tissues. A total of 10 human cartilage samples were collected to verify the RNA sequencing results and we found that circFNDC3B was significantly downregulated in the medial tibial plateau (MTP) compared to the lateral tibial plateau (LTP) (Fig. 1b–d). Then we treated primary human chondrocytes (HCs) with interleukin (IL)-1β and evaluated the circFNDC3B expression levels in these cells. This circRNA was significantly downregulated in IL-1β-induced chondrocytes when compared with the control and this downregulation was shown to be time-dependent (Fig. 1e), suggesting that circFNDC3B may play a key role in IL-1β-induced chondrocytes.

A previous study revealed that the 3′-tail of the exon joins its 5′-head, producing the specialized circular RNA structure associated with circFNBC3B[20]. We went on to design divergent primers to amplify this head-to-tail splicing and confirmed this locus using Sanger sequencing (Fig. 1f). However, trans-splicing or genomic rearrangement may also produce head-to-tail splicing. Therefore, we used a previously described method[20,22] to rule out these possibilities. We designed convergent primers to amplify FNDC3B mRNA and divergent primers to amplify CircFNDC3B using cDNA and genomic DNA (gDNA). CircFNDC3B was amplified when using the divergent primers on the cDNA, but not when using gDNA as the template (Fig. 1g). We also confirmed that circFNDC3B resists RNase R, while FNDC3B mRNA could not resist treatment with RNase R (Fig. 1h). FISH and RT-qPCR indicated that circFNDC3B is primarily expressed in the cytoplasm of human chondrocytes (Fig. 1i, j).

### CircFNDC3B regulates proliferation and ECM metabolism in HCs

To assess the possible functions of CircFNDC3B in regulating matrix-degrading enzymes, we transfected HCs with CircFNDC3B small-interfering RNA (siRNA) (Fig. 2a). SiRNA-circFNDC3B only knocked down the expression of circFNDC3B and had no significant effect on the expression of FNDC3B mRNA (Fig. 2b). Western blotting (WB), RT-qPCR, and immunofluorescence (IF) analysis demonstrated that the knockdown of circFNDC3B increased MMP3 and MMP13 expression while decreasing the levels of Aggrecan and Collagen II (Fig. 2c–e). We then investigated the effect of circFNDC3B on cell proliferation and found that CircFNDC3B knockdown decreased HC proliferation (Fig. 2f).

Furthermore, we refined our evaluations of circFNDC3B regulation in HCs by constructing and then transfecting a circFNDC3B overexpression virus into HCs and then evaluating their response to this upregulation of circFNDC3B expression. RT-PCR showed that this overexpression upregulated the expression of circFNDC3B and had little effect on the expression of FNDC3B mRNA (Fig. 3a, b). To further examine the product of the circFNDC3B overexpression virus, we used RNase R assay. The result showed that the product of the circFNDC3B overexpression virus can resist RNase R, like other circRNAs (Supplementary Figure S2a and b). WB and RT-qPCR showed that IL-1β inhibited the expression of Aggrecan and Collagen II

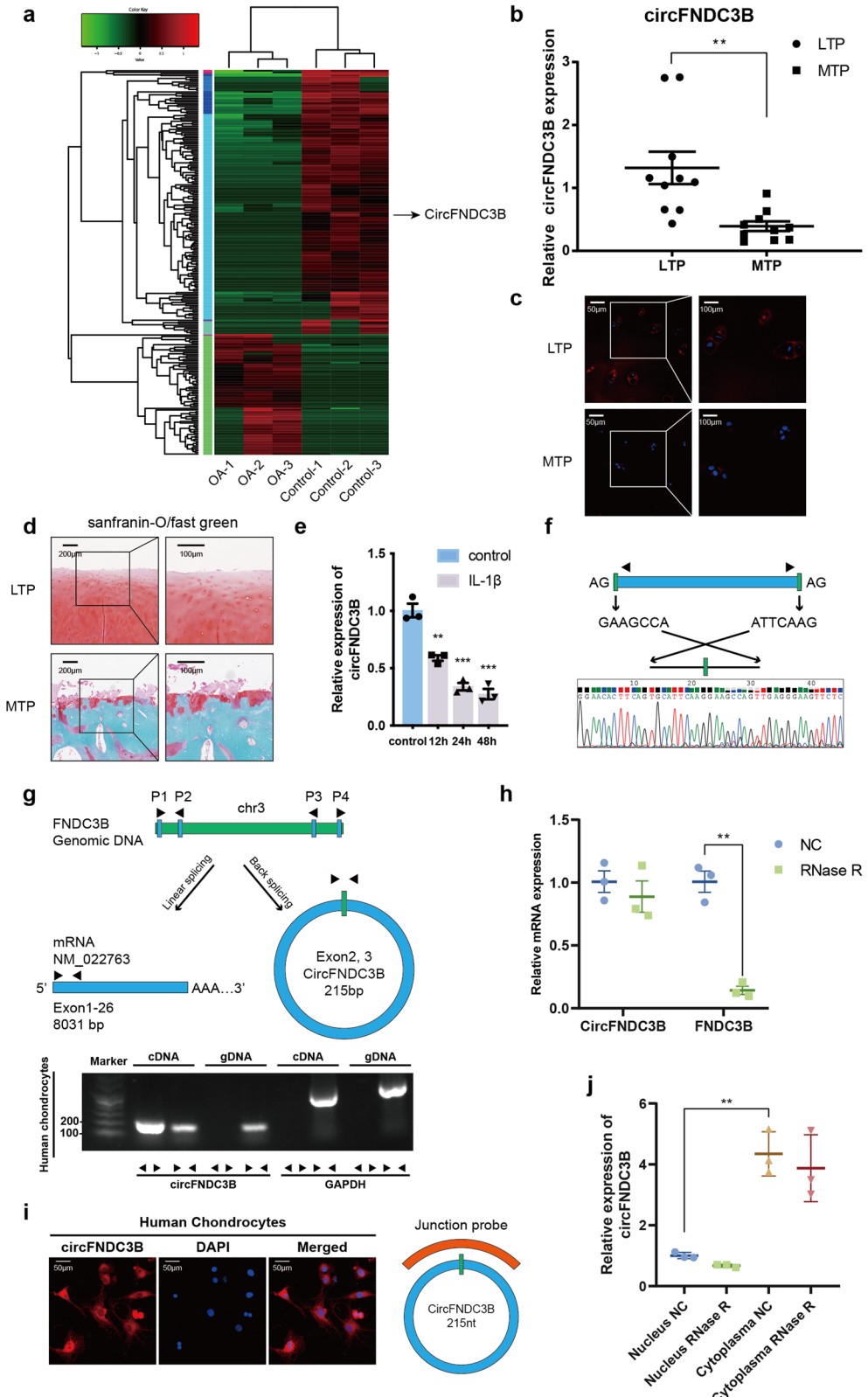

while promoting that of MMP3 and MMP13. However, the overexpression of circFNDC3B antagonized these effects (Fig. 3c, d). In addition, the inhibitory effect of IL-1β on HC proliferation was rescued via the overexpression of circFNDC3B (Fig. 3e). Taken together, these data clearly indicate that circFNDC3B protects HCs by regulating ECM metabolism and cellular proliferation.

**CircFNDC3B sponges miR-525-5p and functions by targeting miR-525-5p.** As CircFNDC3B is primarily expressed in the cytoplasm, we speculated that it functions as an miRNA sponge. A search in the Circular RNA Interactome database revealed that circFNDC3B has conserved Argonaute 2 (AGO2)-binding site. We demonstrated the specific enrichment of endogenous circFNDC3B pulled down from Flag-AGO2 cells by RNA immunoprecipitation (RIP) analysis

**Fig. 1 Characterization of CircFNDC3B in human control and OA tissues. a** Heat map based on OA tissue and control tissue. **b** The relative expression of CircFNDC3B in human medial tibial plateau (MTP) and lateral tibial plateau (LTP) was detected by RT-qPCR. ($n = 10$). **c** The expression of CircFNDC3B in human MTP and LTP was revealed by RNA FISH. Scale bars are 50 μm and 100 μm. **d** Safranin-O/fast green staining of the cartilage from MTP and LTP of human sample. Scale bars are 100 μm and 200 μm. **e** HCs were treated with IL-1β (10 ng/mL) for 12 h, 24 h, and 48 h, the relative CircFNDC3B expression were detected by RT-qPCR. ($n = 3$) **f** The presence of CircFNDC3B was validated by Sanger sequencing. **g** The diagram demonstrated how FNDC3B gene formed FNDC3B mRNA (Left arrow) and circFNDC3B (right arrow). CircFNDC3B was validated by divergent primers. **h** RT-qPCR showed that CircFNDC3B could resist RNase R. ($n = 3$). **i** RNA FISH demonstrated the cellular localization of CircFNDC3B. Scale bar, 50 μm. **j** The expression of CircFNDC3B and FNDC3B mRNA in nucleus or cytoplasm of HCs treated with or without RNase R was detected by RT-PCR. ($n = 3$). Mean ± SEM; **$p < 0.01$; ***$p < 0.001$ by Student's t test.

(Fig. 4a). We identified six candidate miRNAs by overlapping the predicted miRNA recognition elements (MREs) in the circFNDC3B sequence using RNAhybrid, TargetScan, and miRanda (Fig. 4b). The expression levels of all six candidate miRNAs were evaluated by RT-qPCR and miR-525-5p and miR-93-3p were shown to be reasonably highly expressed in HCs and to respond to circFNDC3B-mediated knockdown (Fig. 4c). Furthermore, WB and RT-qPCR results linked miR-525-5p with the expression of several matrix metabolism components and revealed that this association was more significant than that of miR-93-3p (Supplementary Figure S3a and b). We then performed RNA pull-down analysis to validate these candidate miRNAs and found that miR-525-5p was significantly enriched in more than 5% of the input (Fig. 4d). FISH and RT-qPCR results showed that the expression of miR-525-5p in MTP was higher than that in LTP (Fig. 4e, f). Therefore, we chose miR-525-5p for further analyses. Luciferase assay confirmed binding between circFNDC3B and miR-525-5p (Fig. 4g) while the FISH experiments confirmed the co-localization of circFNDC3B and miR-525-5p (Fig. 4h). Given this we went on to investigate the effects of miR-525-5p in HCs. Over-expression of miR-525-5p increased the expression of MMP3 and MMP13, and decreased the expression of Collagen II and Aggrecan, as determined by RT-qPCR (Fig. 5a) and WB (Fig. 5c). As expected, miR-525-5p knockdown exerted the opposite effects on matrix metabolism components (Fig. 5b, c). To investigate whether circFNDC3B functions by targeting miR-525-5p in OA, we co-infected HCs with si-circFNDC3B and miR-525-5p inhibitor. WB and qRT-PCR analyses demonstrated that the downregulation of miR-525-5p antagonized the effect of circFNDC3B knockdown on MMP3, MMP13, Collagen II, and Aggrecan in HCs (Fig. 5d, e). In addition, the downregulation of miR-525-5p antagonized the effect of circFNDC3B knockdown on the rate of HC proliferation (Fig. 5f). Taken together, these results suggest that circFNDC3B functions by sponging miR-525-5p in vitro.

**MiR-525-5p directly targets heme oxygenase 1.** We then used PubMed to identify all the OA-related genes and then overlapped them with the predicted target sites for miR-525-5p produced using Targetscan, miRDB, and PITA. This evaluation identified eight potential targets, including SPP1, HO-1, APLN, PTEN, BAX, FN1, TNFRSF1A, and OLR1 (Fig. 6a). RT-qPCR showed that HO-1 (heme oxygenase 1) was significantly regulated by circFNDC3B (Fig. 6b) and we used TargetScan to predict the putative miRNA binding sites in the 3′-UTR of HO-1 mRNA (Fig. 6c). We hypothesized that miR-525-5p might exert its functions by regulating HO-1 expression in HCs. Therefore, we used a luciferase activity assay to investigate the relationship between miR-525-5p and HO-1. The results demonstrated that miR-525-5p overexpression significantly downregulates the luci-ferase activity of the reporter gene in wild-type constructs, but not in mutant constructs (Fig. 6d). WB and qRT-PCR results showed that the knockdown of circFNDC3B and overexpression of miR-525-5p decreased the expression of HO-1 (Fig. 6e, f). Therefore, HO-1 was chosen for further analyses. RT-qPCR and IF analysis showed that the expression of HO-1 in LTP was higher than that

in MTP (Fig. 7a, b). Therefore, HO-1 was confirmed as an important downstream target for further research. WB and RT-qPCR showed that HO-1 knockdown decreased the expression of HO-1, Collagen II, and Aggrecan, and increased that of MMP3 and MMP13 (Fig. 7c, d). In contrast, overexpression of HO-1 had the opposite effects on these matrix-degrading and synthesizing components (Fig. 7e, f). Moreover, the knockdown of circFNDC3B and overexpression of miR-525-5p decreased the expression of HO-1, Collagen II, and Aggrecan, and increased that of MMP3 and MMP13, while the overexpression of HO-1 antagonized the effect of circFNDC3B knockdown and miR-525-5p overexpression (Fig. 7g, h). Taken together, these results suggest that the circFNDC3B/miR-525-5p axis functions to reg-ulate HO-1, and HO-1 exhibits a similar function to circFNDC3B in chondrocytes.

**Oxidative stress and HO-1/ NF-κB pathway mediates the CircFNDC3B/miR-525-5p/HO-1 axis in OA.** Oxidative stress is known to be detrimental to many cells and occurs during disease and aging. It has also been implicated in the development of OA[23]. Thus, it is essential to investigate the effects of antioxidants on the pathogenesis of OA. In a recent study by Takada et al., the authors concluded that an increased level of HO-1 may protect against OA development in both aging and post-traumatic OA[24]. In addition, p-P65 and p-P50 levels were lower in HO-1-overexpressing cells, suggesting an inhibition of the NF-κB-mediated effects in these evaluations[16]. As HO-1 is the target of our study, we performed RT-qPCR and WB to determine whe-ther the circFNDC3B/miR-525-5p/HO-1 axis interacts with the NF-κB pathway. Figure 8a–c show that the knockdown of circFNDC3B decreased the expression of HO-1 and increased that of p-P65 and p-P50. In contrast, overexpression of HO-1 or the knockdown of miR-525-5p had the opposite effect. Intracel-lular ROS was measured using a BD FACS Calibur flow cyt-ometer following DCFH-DA labeling in each group (Fig. 8d). The results showed that circFNDC3B knockdown significantly upre-gulated ROS in the chondrocytes, while both the knockdown of miR-525-5p and the overexpression of HO-1 attenuated ROS levels in these cells. These results indicate that the CircFNDC3B/miR-525-5p/HO-1 axis interacts with the NF-κB pathway and influences the intracellular ROS level in chondrocytes.

**CircFNDC3B alleviates OA in a rabbit ACLT model.** To verify whether CircFNDC3B functions in OA progression in vivo, wild-type (WT) or mutant (MUT) adeno-associated virus (AAV) circFNDC3B was intra-articularly administered into anterior cruciate ligament transection (ACLT)-induced OA rabbits. RT-qPCR were performed to evaluate the overexpression levels of CircFNDC3B (Fig. 9a). Safranin O and fast green staining showed the substantially thickened cartilage layer on the articular surface after the injection of WT AAV CircFNDC3B; however, the injection of MUT AAV circFNDC3B did not have this effect (OA + MUT group) (Fig. 9b). In addition, WT AAV circFNDC3B significantly reduced OARSI scores, while there was

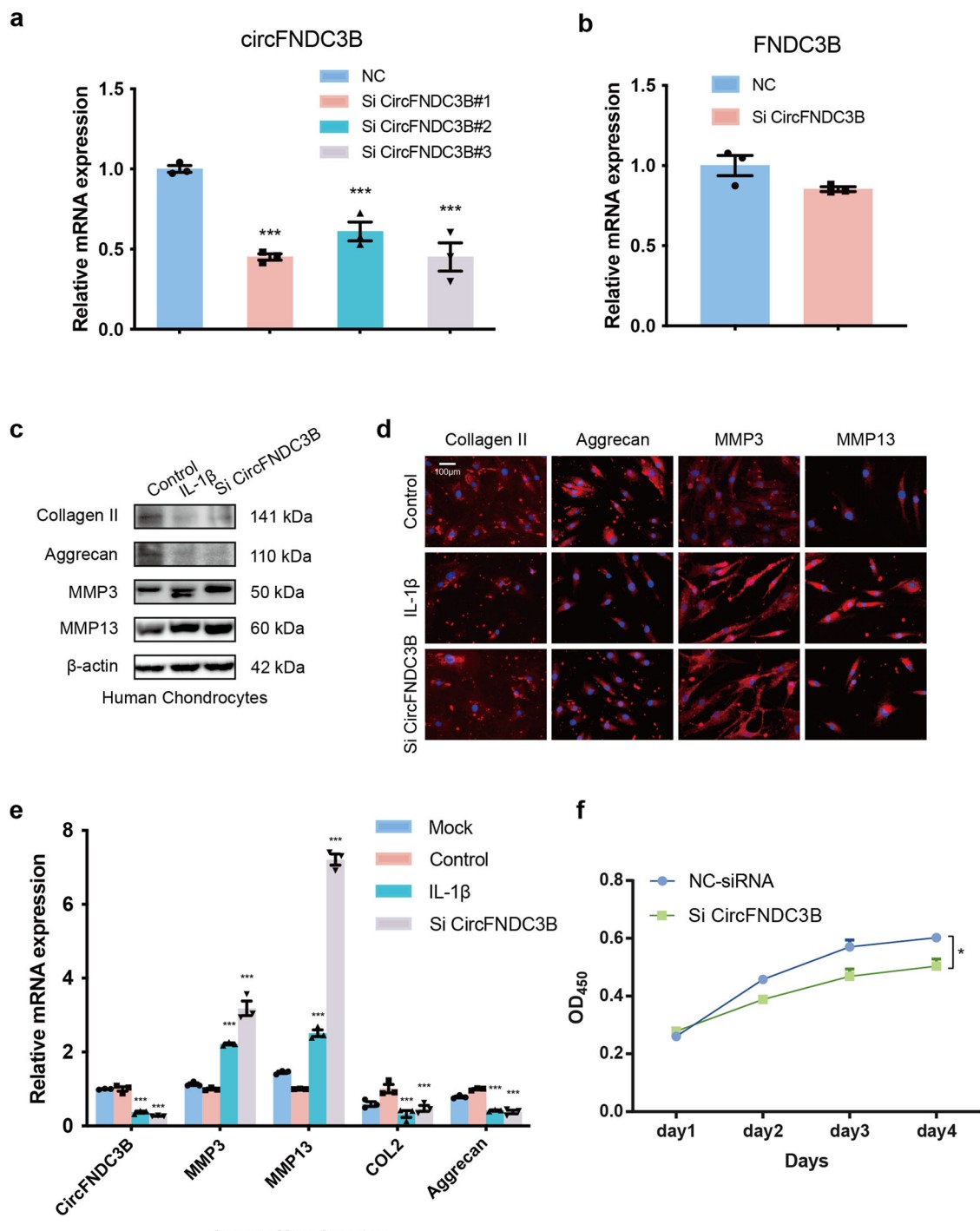

**Fig. 2 Knock-down of CircFNDC3B inhibits proliferation and ECM metabolism in chondrocytes. a, b** RT-qPCR analysis of the expression levels of CircFNDC3B and FNDC3B mRNA after CircFNDC3B knockdown. ($n = 3$). ***$p < 0.001$ by Student's t test. **c, d** Western blot analysis and IF (Immunofluorescence) of Collagen II, Aggrecan, MMP3, and MMP13 when HCs were treated with IL-1β or siRNA, β-actin was used as a loading control. Scale bar, 100 μm. **e** RT-PCR analysis of Collagen II, Aggrecan, MMP3, and MMP13 when HCs were treated with nothing, IL-1β, NC-siRNA or siRNA. ($n = 3$). ***$p < 0.001$ by Student's t test. **f**. HCs viability was detected by the CCK-8 assay. ($n = 3$). *$p < 0.05$ by ANOVA.

no change in the OARSI scores of the MUT AAV circFNDC3B group (Fig. 9c). Immunohistochemistry (IHC) and WB revealed that treatment with circFNDC3B decreased the expression of MMP3 and MMP13 and increased HO-1, Collagen II, and Aggrecan levels in chondrocytes. This implied that circFNDC3B alleviated the degenerative changes in the cartilage matrix and increased ECM composition in the rabbit OA model (Fig. 9d, e).

Together, these results demonstrate the protective role of circFNDC3B in OA in vivo (Fig. 9f).

## Discussion

OA is one of the most common chronic diseases, with the World Health Organization estimating the prevalence of symptomatic

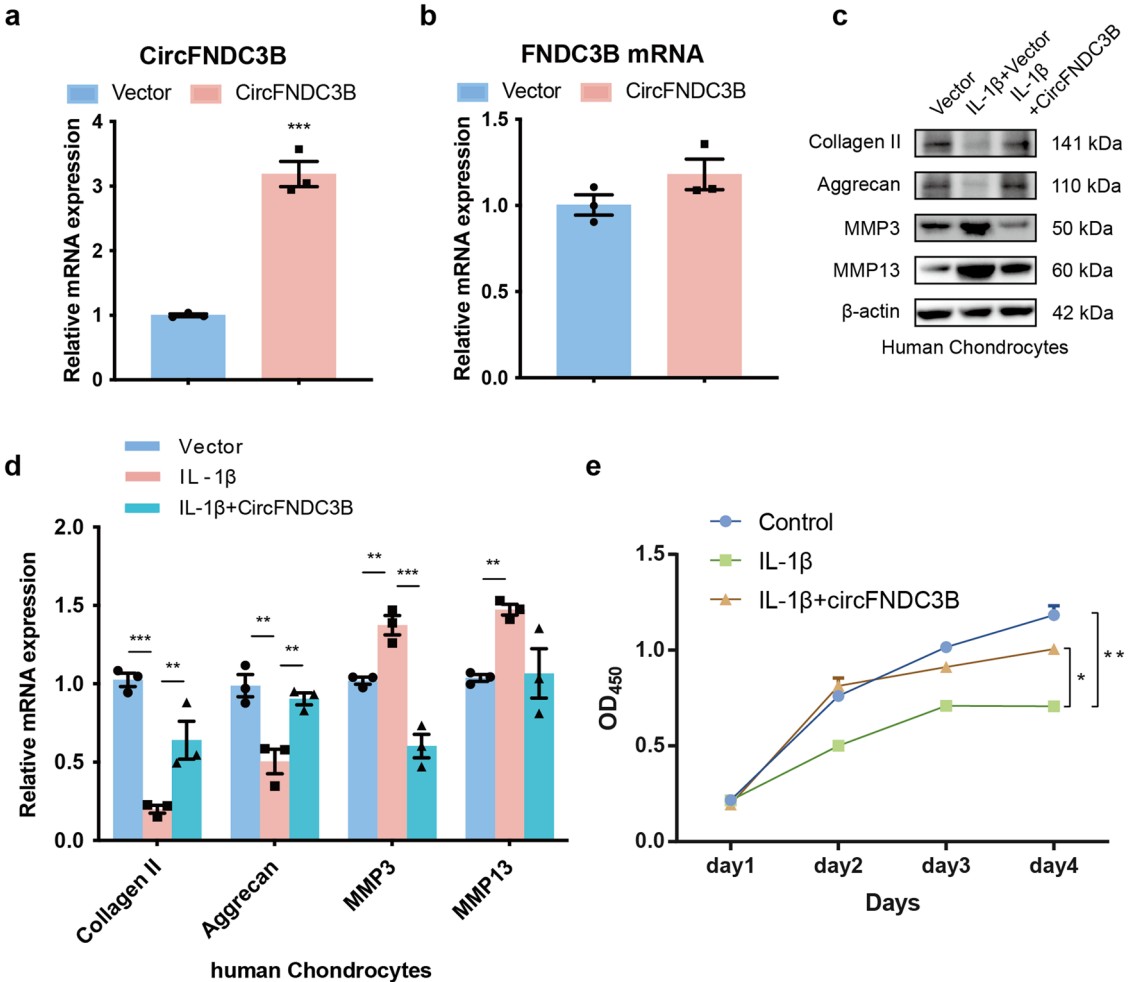

**Fig. 3 Overexpression of CircFNDC3B promotes proliferation and ECM metabolism in HCs. a** RT-qPCR analysis of the overexpression efficiency of CircFNDC3B in HCs. Vector (empty vector) was used as control. ($n = 3$). ***$p < 0.001$ by Student's t test. **b** RT-qPCR analysis of the relative level of FNDC3B mRNA in the HCs after overexpression of CircFNDC3B. ($n = 3$). **c, d** Western blot and RT-qPCR analysis of Collagen II, Aggrecan, MMP3, and MMP13 when HCs were treated with IL-1β and the rescuing effects of overexpressed CircFNDC3B on IL-1β. β-actin was used as a loading control. ($n = 3$). **$p < 0.01$, ***$p < 0.001$ by Student's t test. **e** HCs viability was detected by the CCK-8 assay. ($n = 3$). *$p < 0.05$, **$p < 0.01$ by ANOVA.

OA, among people over 60 years of age, at 9.6% in men and 18.0% in women[10]. Despite this high prevalence, efficient treatments for slowing OA progression are still not available. Current therapeutic strategies for OA, such as joint replacement surgery and drugs, focus on relieving the symptoms of the disease rather than curing it[25]. This means that better elucidation of the underlying molecular mechanisms driving OA pathogenesis is urgently needed.

Oxidative stress is elevated in the joint tissues, especially in the cartilage, during OA progression and the aging process[26]. Several studies have provided evidence for the use of antioxidants in reducing OA severity, but their underlying mechanism remains unknown. In addition, relatively few studies have focused on the relationship between circRNAs, oxidative stress, and OA. Therefore, this study was designed to evaluate the cellular mechanisms underlying circRNA-mediated changes in oxidative stress and their association with OA in the hope of providing novel insights into the treatment of OA. To our knowledge, this is the report of a regulatory role of hsa_circ_0001361 (also called circFNDC3B) in OA. We conducted a series of analyses which demonstrated that circFNDC3B acted to suppress oxidative stress, promote chondrocyte proliferation, and protect the ECM from degradation.

CircRNAs are characterized by a closed-loop structure, which improves their stability[20] and several recent studies have discovered that these circRNAs often function as miRNA sponges[19,21,27]. Our evaluations revealed that circFNDC3B retains a closed structure and functions as a sponge for miR-525-5p. Expression of circFNDC3B has been linked to bladder cancer[28], hepatocellular carcinoma[29] and neuroblastoma[30]. Here, we identified a novel regulatory axis in HCs, namely that of circFNDC3B/miR-525-5p/HO-1, which may be activated to relieve OA. HO-1 has a known effect against osteoporosis[31], gastric cancer[32], myocardial infarction[33], and Lupus Nephritis[34]. Targetscan, miRDB, and PITA databases all predicted that HO-1 acts as a target of miR-525-5p, and its importance was demonstrated in OA. Recent studies have underlined the importance of HO-1 in inflammation[35] and age-related diseases[36] where it acts as a critical element of the antioxidant response. It has also been reported to target oxidative stress and reduce OA[10]. In addition, increases in HO-1 activity lead to the inhibition of the NF-κB pathway[37]. The NF-κB pathway is widely known to be an important inflammatory regulator involved in the development of various diseases including OA. It also plays an important role in cartilage degradation[38], and induces a variety of inflammation-related factors, including several MMP proteins (in our study,

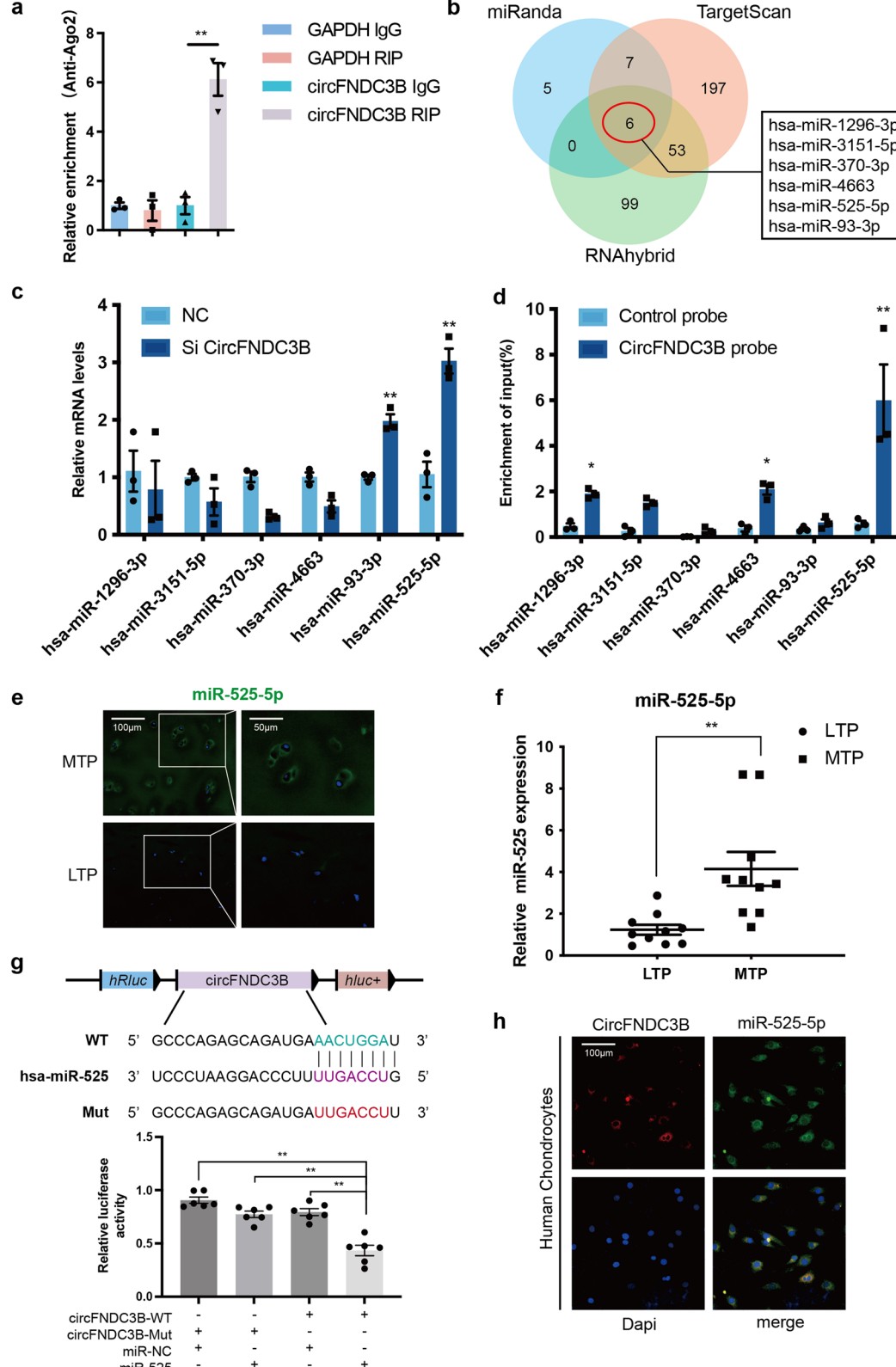

MMP3 and MMP13), inducible nitric oxide synthase (iNOS), interleukin-1β (IL-1β), and tumor necrosis factor-α (TNF-α). MMP13 degrades collagen II and aggrecan by cleaving the helical protein[39], while ADAMTS-5 degrades aggrecan. This is significant as these are the two most abundant components in the ECM. Our findings indicate that HO-1 was downregulated in the joint-wearing zone and was regulated by both circFNDC3B and

miR-525-5p. Moreover, as an important downstream target of the circFNDC3B/miR-525-5p axis, HO-1 overexpression was shown to play a protective role in OA.

The conservation of circFNDC3B among higher species may suggest a more important role of this circRNA in OA progression. Further, given the fact that we found the sequence of circFNDC3B to be relatively conserved between humans and

**Fig. 4 CircFNDC3B serves as sponge for miR-525-5p. a** Quantitative real-time PCR (RT-qPCR) quantification of argonaute-2 (AGO2)-bound CircFNDC3B, immunoprecipitated (IP)/ input values normalized to negative control (NC). ($n = 3$). **b** The target miRNAs of CircFNDC3B were predicted by miRanda, Targetscan, and RNAhybrid. The diagram demonstrated the overlapping results. **c**. RT-qPCR analysis of the relative level of 6 miRNA candidates in the HCs lysates. ($n = 3$). **d** The CircFNDC3B-binding miRNAs were detected by RNA-pull-down analysis and RT-PCR quantification. ($n = 3$). **e** Representative images of miR-525-5p (green) labeled FISH staining in MTP and LTP. Scale bars, 100 μm, 50 μm. **f** RT-qPCR analysis of the relative level of miR-525-5p in the MTP and LTP. ($n = 3$). **g** Upper panel, diagram demonstrated complementary to the miR-525-5p seed sequence with CircFNDC3B. Lowercase letters indicate mutated nucleotides. Lower panel, relative luciferase activities were measured after HEK-293T cells were co-transfected with miR-525-5p mimics and luciferase reporter vector. ($n = 3$). **h** RNA FISH images demonstrated the co-localization of CircFNDC3B (red) and miR-525-5p (green) in HCs. Scale bar, 100μm. Mean ± SEM; *$p < 0.05$; **$p < 0.01$; ***$p < 0.001$ by Student's t test.

rabbits, it is possible to explore the significance of circFNDC3B in rabbit models of joint disease. CircFNDC3B was upregulated in a rabbit model when treated with a circFNDC3B AAV and its overexpression alleviated ACLT-induced OA. These results confirmed our hypothesis that circFNDC3B may play a protective role in OA.

However, our study has some limitations. CircRNAs are generally studied because they regulate downstream gene expression. However, their biogenesis is poorly understood. A previous study showed that the RNA-binding protein FUS can affect circular RNA expression[40–42]. Our study could be improved by evaluating the upstream mechanisms of the circFNDC3B/miR-525-5p/HO-1 axis. FUS primarily regulates circRNA biogenesis through two GUGGU-binding motifs[43]. In our study, circFNDC3B was constructed from exons 2 and 3 of FNDC3B and we found one GUGGU sequence at position 701 before exon 2 and one at 111 nt after exon 3, making it feasible that circFNDC3B biogenesis is regulated by FUS/TLS. Therefore, further studies evaluating the upstream regulation of the circFNDC3B/miR-525-5p/HO-1 axis are necessary and may be critical to our understanding of this pathway. In addition, due to the 3 R principle to spare animals, we only set the sham surgery group, rather than set 3 control group (sham+NC, sham+circFNDC3B,sham+circFNDC3B mut). Moreover, the effect of circFNDC3B was only been studied in cartilage. CircFNDC3B can be facilitate secretion into the synovial fluid. It is rational to infer that circFNDC3B may play a role in synovial fluid and synovial membrane. In summary, to our knowledge, our study is the first to identify a novel signaling pathway, circFNDC3B/miR-525-5p/HO-1, which could relieve OA by alleviating oxidative stress and regulating the NF-κB pathway, leading to ECM protection in human chondrocytes. CircFNDC3B also exhibits strong sequence conservation across species making it an ideal target for the development of novel therapies for OA, as the efficacies of these therapies can be efficiently verified via various in vivo models of OA. The results of this study posit circFNDC3B as a potential target for the development of novel effective therapies for OA.

## Materials and methods

**Human cartilage and chondrocytes**. Collections of human cartilage samples were according to protocols approved by the Ethics Committee of Sir Run Run Shaw Hospital (Hangzhou, China), and the methods were carried out in accordance with the approved guidelines. All subjects signed a written informed consent. Human cartilage samples were obtained from patients who underwent total knee replacement surgery ($n = 10$). Patient exclusion criteria are detailed in reference. [21]. The knee was an important target site for OA, the medial tibiofemoral joint was most affected, and isolated lateral tibiofemoral joint OA was relatively rare[44]. The lesions on the medial tibial plateau (MTP) were more evident than the lateral tibial plateau (LTP). Hence, these two areas of cartilage tissues were collected for subsequent analysis. Human chondrocytes and rabbit chondrocytes were harvested from human and rabbit cartilage. The details have been reported in reference. [21]. Chondrocytes were maintained in DMEM containing 10% fetal bovine serum (FBS; Thermo Fisher Scientific, Waltham, MA, USA) for 24 h at 37 °C. The cells were filtered through a 0.075 mm cell strainer and washed with sterile phosphate buffered saline (PBS) before culturing or miRNA/mRNA isolation. Primary chondrocytes at 80% confluence were used for the experiments. During the culture

period, the cells were incubated at 37 °C in a humidified atmosphere of 5% $CO_2$ and 95% air.

**A rabbit model of osteoarthritis**. All rabbits (12-month-old male New Zealand white rabbits, weighing 2.5-3 kg) were purchased from Xin Jian rabbit field (Certificate No. SCXK, Zhejiang, 2015-0004, China). A total of 32 rabbits were used in vivo experiments. As reported by Yoshioka et al[45], anterior cruciate ligament transection (ACLT) surgery was performed to induce post-traumatic osteoarthritis model. In short, a medial parapatellar incision was conducted and an arthrotomy was operated. The patella was dislocated laterally, and the knee completely flexed. The Anterior Cruciate Ligament (ACL) was exposed and transected with a No.12 blade (supplementary figure S1a and b). To confirm that the ACL surgery, we performed the anterior drawing test after each procedure. The joint was washed with sterile saline and then closed. The bilateral knee joint was performed ACLT surgery. After the operation, the rabbits could move freely in the cage without immobilization. 24 rabbits underwent the ACLT operation and the remaining 8 underwent sham operation (as control group). The c virus (AAV) CircFNDC3B WT and Mut were purchased from HanBio (Shanghai, China). All 32 rabbits were randomly distributed into 4 groups of 8 rabbits per group (shame surgery group with saline injection, ACLT surgery group with saline injection, ACLT surgery group with CircFNDC3B WT virus injection, ACLT surgery group with CircFNDC3B Mut virus injection), and each rabbit was raised in a single cage. After the surgery, the rabbits were rested for a week. Then the knee joint cavities of rabbits were injected with a total of 100 μl saline or virus solution (approximately $1 \times 10^8$ PFU/mL). After 7 weeks, all rabbits were sacrificed, and the knee joints were isolated for further research. The timeline of the whole rabbit experiment was visualized (supplementary figure S1c). The rearing and experiments were performed strictly with the approval of the Institute of Health Sciences Institutional Animal Care and Use Committee (Zhejiang, China).

**Safranin O-fast green staining and OARSI score**. Cartilage specimens were fixed and decalcified, then sectioned at 5 μm, and each 10th section was stained with Sigma-Aldrich safranin O solution and Fast Green solution (St. Louis, MO, USA). The Osteoarthritis Research Society International (OARSI) score was based on safranin O-fast green staining[19], the details were reported in reference. [21].

**RNA pull-down assay with biotinylated CircFNDC3B probe**. This assay was performed according to the instrument using the RNA pulldown kit (BersinBio, Guangzhou, China) and the biotinylated CircFNDC3B probe was designed and synthesized by RiboBIO (Guangzhou, China). The details have been reported in reference. [46].Final RNA samples were subjected by RT-qPCR for detection.

**Bioinformatics analysis**. TargetScan, RNAhybrid and miRanda[47–49] were used to predict the target of circRNA. TargetScan, miRDB and PITA were used to predict the target of miRNA as described[19]. Osteoarthritis-related genes were obtained from NCBI Gene database.

**RNA extraction and quantitative real-time PCR analysis**. Total cellular RNA was extracted from the cultured chondrocytes using TRIzol reagent (Invitrogen, Carlsbad, CA, USA), according to the manufacturer's instructions. miRNA levels were extracted using miRNA Isolation Kit (Ambion). RNA was stored at −80 °C. Reverse transcription was performed using 1.0 μg total RNA and a miRNA cDNA Kit or HiFiScript cDNA Kit (CWBIO, Beijing, China), which were used to investigate the expression of miRNA and mRNA, respectively. Quantification real-time PCR was performed using the Hieff qPCR SYBR Green Master Mix (Yeason Biotech, China) and analyzed on an ABI 7500 Sequencing Detection System (Applied Biosystems, Foster City, CA, USA). All involved primers are listed in Supplementary Table S1.

**RNA knockdown and overexpression**. Small interfering RNAs (siRNAs) targeting the CircRNA (CircFNDC3B) or mRNA (HO-1), as well as the mimic or inhibitor of miRNA (miR-525-5p) were purchased from Ribobio (Guangzhou, China).

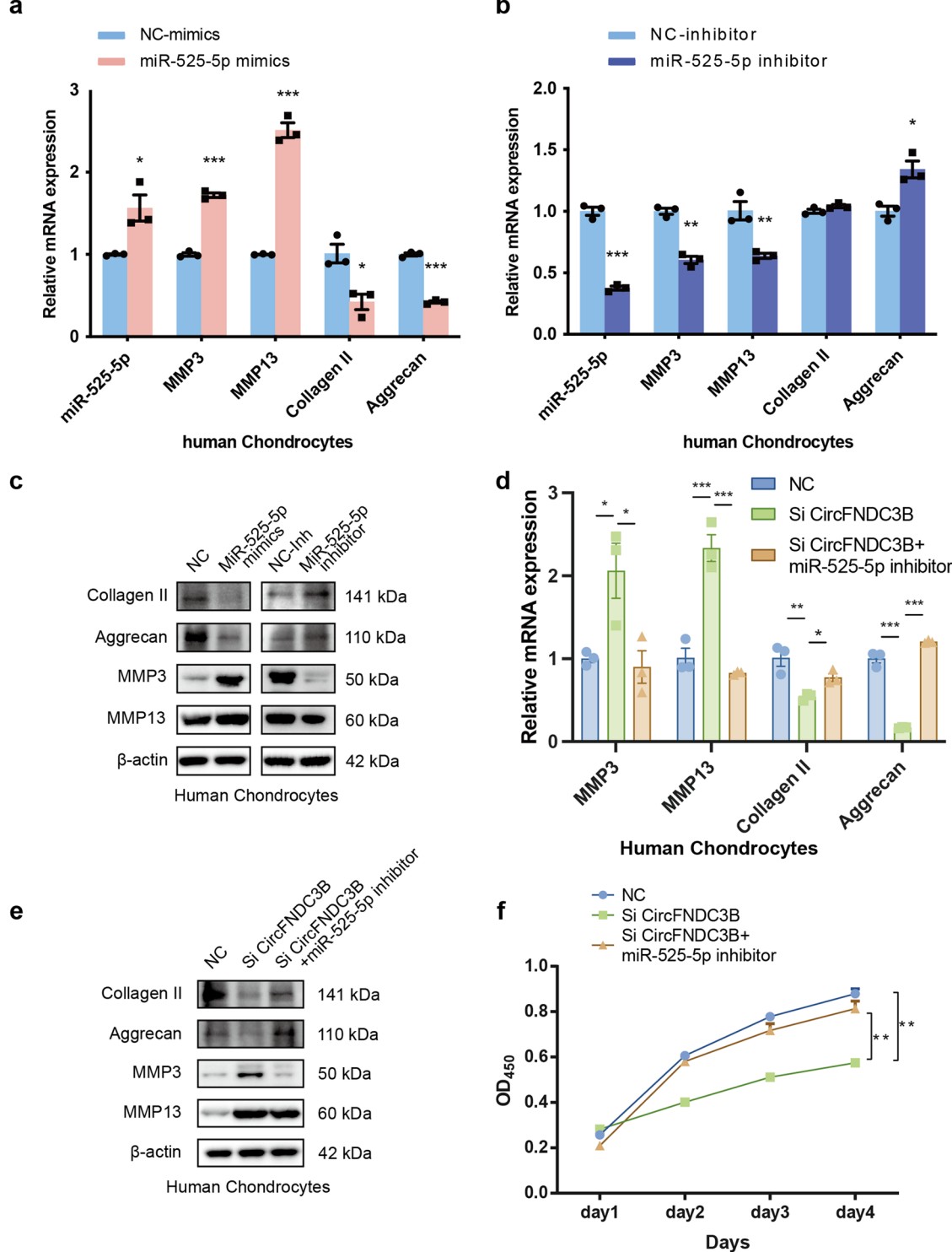

**Fig. 5 CircFNDC3B functions by targeting miR-525-5p. a, b** RT-qPCR analysis of Collagen II, Aggrecan, MMP3, and MMP13 when HCs were overexpression or knockdown of miR-525-5p. ($n = 3$). *$p < 0.05$, **$p < 0.01$, ***$p < 0.001$ by Student's t test. **c** Western blot analysis of Collagen II, Aggrecan, MMP3, and MMP13 when HCs were overexpression or knockdown of miR-525-5p. **d–f** RT-PCR, western blot and CCK-8 assay showed that the downregulation of miR-525-5p antagonized the effect of si-CircFNDC3B on Collagen II, Aggrecan, MMP3, and MMP13 in HCs. ($n = 3$). RT-PCR results: *$p < 0.05$, **$p < 0.01$, ***$p < 0.001$ by Student's t test. CCK-8 assay results: **$p < 0.01$ by ANOVA.

Overexpression vector for CircFNDC3B was constructed by Tsingke (Beijing, China) using the pc016:pcDNA3.1-CMV-circRNA-Zsgreen plasmid vector. Cells transfection with plasmids was accomplished using Lipofectamine 3000 transfection reagent (ThermoFisher). SiRNAs, inhibitors and mimics were transfected using Lipofectamine RNAiMAX transfection reagent (ThermoFisher).

**Dual-luciferase reporter assay**. Luciferase reporter assay was used to determine the binding between circRNA and miRNA, or between miRNA and mRNA. The luciferase reporter vectors (plasmid vector: hFLuc-XbaI-hRLuc) were designed and purchased from Genechem (Shanghai, China). The 3′UTR sequence of CircFNDC3B or HO-1 and their mutants were inserted into XbaI restriction sites

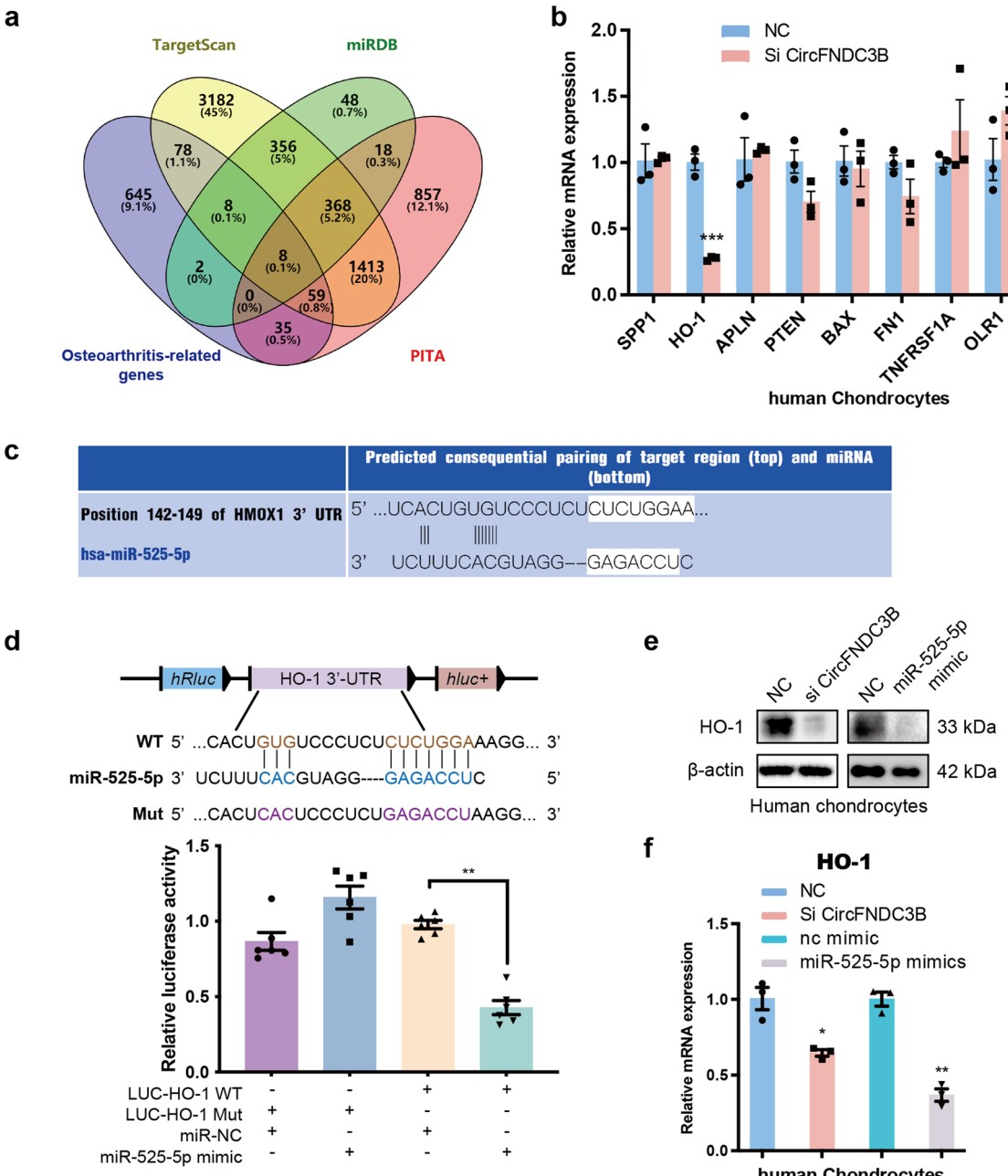

**Fig. 6 MiR-525-5p directly targets heme oxygenase 1. a** The target mRNAs of miR-525-5p were predicted by miRDB, Targetscan, PITA and Osteoarthritis-related genes. The diagram demonstrated the overlapping results. **b** RT-qPCR analysis of 8 mRNA candidates in the HCs lysates. ($n = 3$). **c** Schematic illustration showing the predicted binding region of miR-525-5p and HO-1. **d** Upper panel, schematic illustration demonstrates complementary to the miR-525-5p seed sequence with HO-1. Lowercase letters indicate mutated nucleotides. Lower panel, HEK-293T cells were co-transfected with miR-145-5p mimics and a luciferase reporter construct containing wild-type (WT) or mutated HO-1. ($n = 3$). **e, f** Western blot and RT-qPCR analysis of HO-1 in HCs after treated with si-CircFNDC3B or miR-525-5p mimic. ($n = 3$). Mean ± SEM; *$p < 0.05$; **$p < 0.01$; ***$p < 0.001$ by Student's t test.

of the luciferase reporter vectors. The detailed description were reported in reference. [50]. Experiments were independently repeated three times.

**Western blotting**. Human or rabbit chondrocytes were lysed with RIPA buffer (Beyotime, China), protein was collected, protein concentration was determined by BCA analysis (Beyotime, China). SDS-PAGE separated total cellular protein, which was then transferred to an Immobilon-P membrane (0.2 μm pore size, Millipore, Billerica, MA, USA). The membranes were blocked and then incubated with primary antibodies (BD Biosciences, San Jose, CA, USA) overnight at 4 ℃, and subsequently incubated with secondary antibodies for 1 h. Imaging was conducted with FDbio-Femto ECL (Fudebio, Hangzhou, China) and a chemiluminescence system (Bio-Rad, USA), analysis was performed using Image Lab Software. The antibodies we used were listed in Supplementary Table S2. Especially, according to

the manufacturing company, the MMP13 recognition site is synthetic peptide within Human MMP13 (C terminal), and the MMP3 recognition site is synthetic peptide within Human MMP3 aa 450 to the C-terminus (C terminal). Experiments were independently repeated three times.

**Immunofluorescence**. Cells were fixed with 4% paraformaldehyde for 30 min, permeated with 0.5% tritonX-100 for 30 min and blocked with 5% bovine serum albumin (BSA) for 1 h. The cells were then incubated with primary antibodies (diluted 1:100 by BSA) at 4 ℃ overnight, washed thrice with PBS and incubated with CL594- or CL488-conjugated secondary antibodies (Proteintech Group, Rosemount, IL, USA, diluted 1:200 by BSA) at room temperature for 1 h. Finally, after washing thrice with PBS, the nuclei were stained with DAPI. All operations, starting with the incubation of secondary antibodies, were performed in the dark.

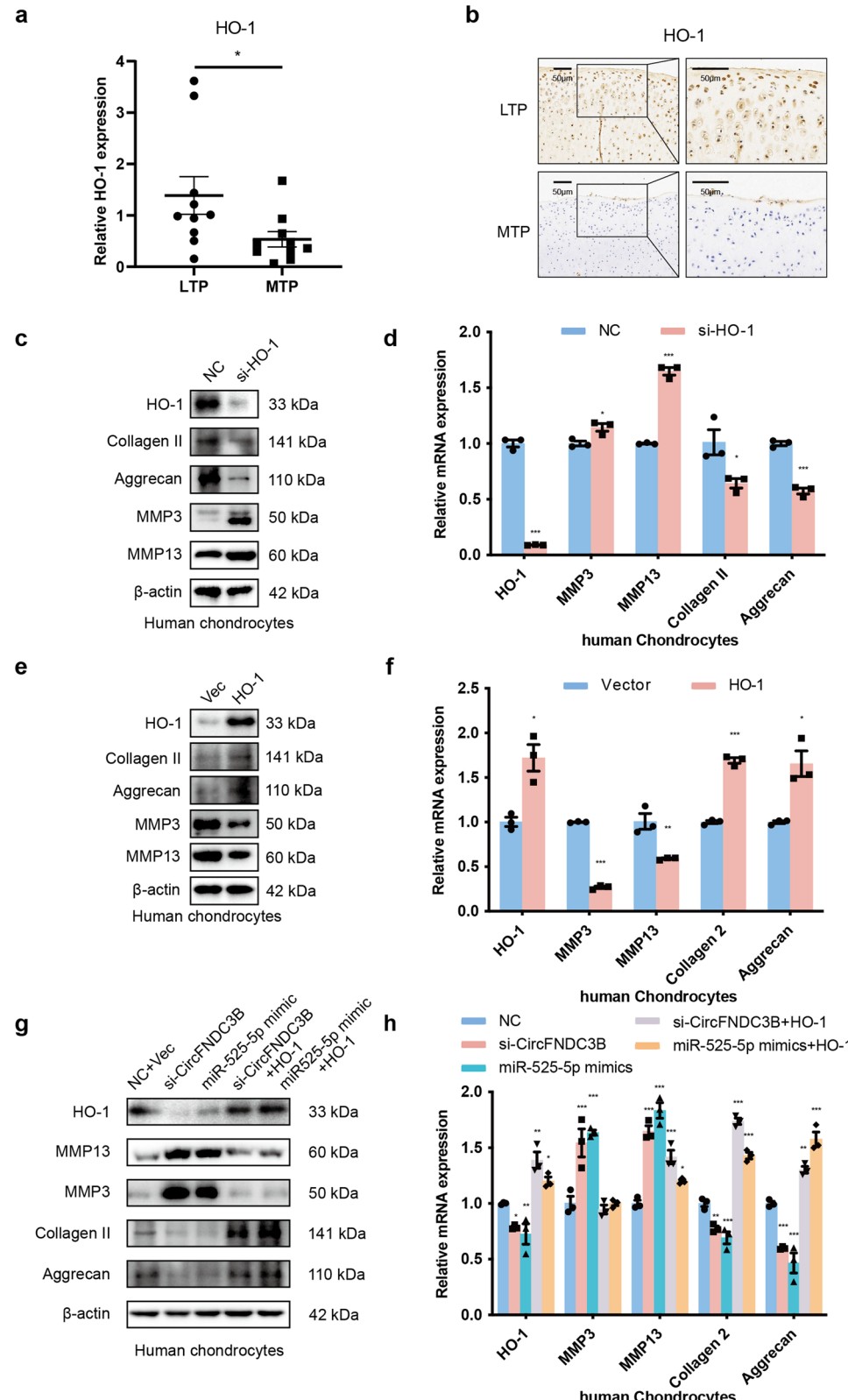

**Fig. 7 HO-1 is down-regulated in OA tissue and confirmed as the downstream gene for CircFNDC3B and miR-525-5p. a** RT-qPCR analysis of HO-1 in different stress areas of 10 human cartilage samples. (*n* = 10). **b** Representative images of immunohistochemistry staining of HO-1 in MTP and LTP. Scale bar, 50 μm. **c**, **d** Western blot and RT-PCR analysis of HO-1, MMP3, MMP13, Collagen II, and Aggrecan when HO-1 was downregulated in HCs. (*n* = 3). **e**, **f** Western blot and RT-PCR analysis of HO-1, MMP3, MMP13, Collagen II, and Aggrecan when HO-1 was overexpressed in HCs. (*n* = 3). **g**, **h** Western blot and RT-PCR analysis showed that overexpression of HO-1 could antagonize the effects of si-CircFNDC3B and miR-525-5p mimic on HO-1, MMP3, MMP13, Collagen II, and Aggrecan in HCs. (*n* = 3). Mean ± SEM; *$p < 0.05$; **$p < 0.01$; ***$p < 0.001$ by Student's t test.

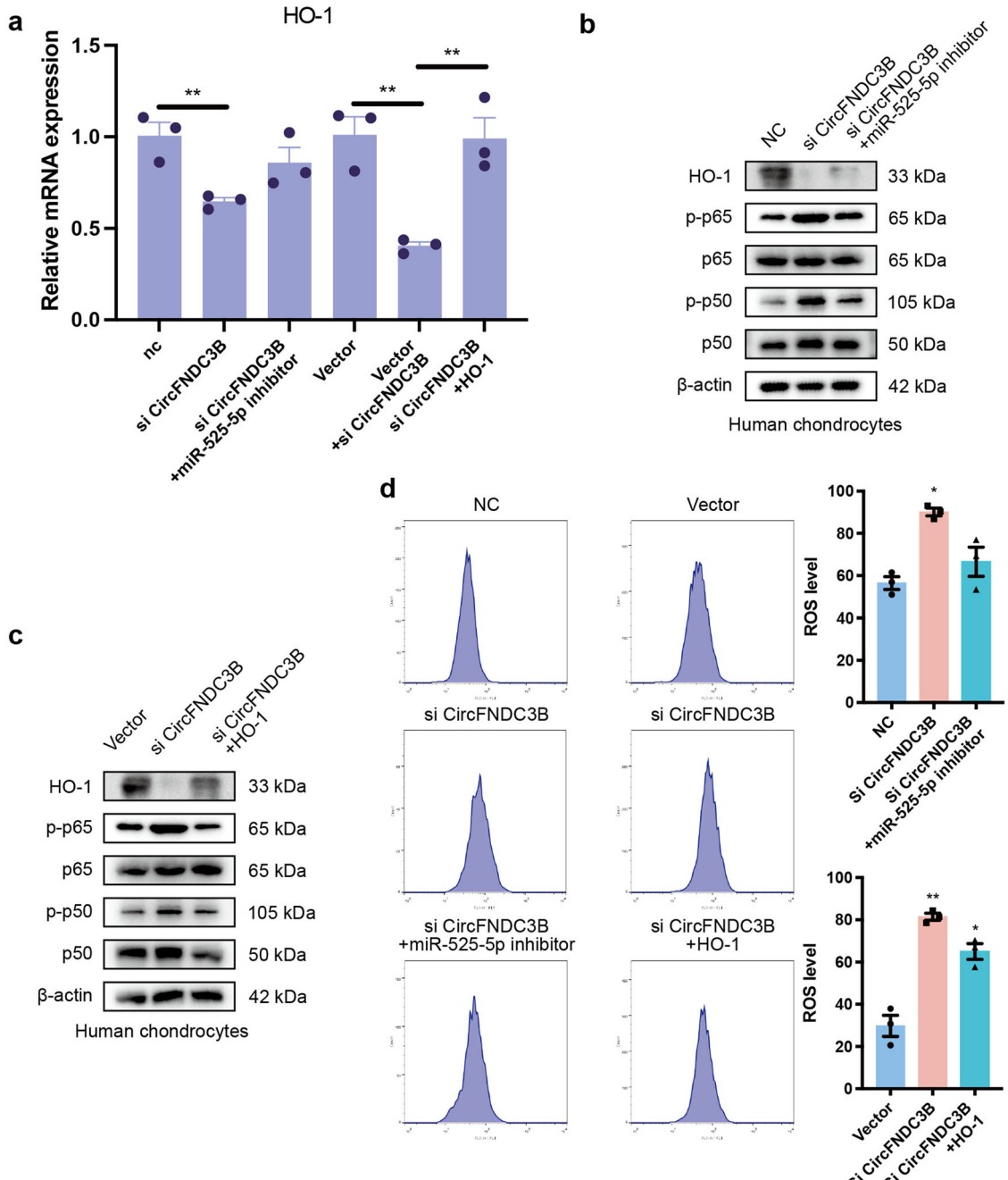

**Fig. 8 Oxidative stress and HO-1/ NF-κB pathway mediates the CircFNDC3B/miR-525-5p/HO-1 axis in OA. a**. RT-PCR analysis showed that the downregulation of miR-525-5p or the overexpression of HO-1 antagonized the effect of si-CircFNDC3B on HO-1 in HCs. Vector (empty vector) was used as control. ($n = 3$). **b, c**. Western blot analysis showed that the downregulation of miR-525-5p or the overexpression of HO-1 antagonized the effect of si-circFNDC3B on p65, p-p65, p50, p-p50 and HO-1 in HCs. **d**. HCs were stained with DCFH-DA, and their oxidative stress was measured by BD FACS Calibur flow cytometer. ($n = 3$). Mean ± SEM; *$p < 0.05$; **$p < 0.01$; ***$p < 0.001$ by Student's t test.

All images were acquired using a fluorescence microscope (Eclipse E600; Nikon Corporation, Tokyo, Japan). The antibodies we used were listed in Supplementary Table S2.

**Immunohistochemistry**. Cartilage specimens were fixed in 4% paraformaldehyde for paraffin embedding and sectioned at 5μm. The sections were incubated with primary antibody at 4 °C overnight, washed thrice with PBST and incubated with a secondary antibody (Beyotime Institute of Biotechnology, Inc., Jiangsu, China) for 2 h at room temperature.

**RNA fluorescent in situ hybridization (FISH)**. The FISH assay was conducted in HCs or rabbit tissues. Cy3-labeled CircFNDC3B probes and 488-labeled locked nucleic acid miR-525-5p probes were designed and synthesized by RiboBio

(Guangzhou, China). RiboBio FISH Kit (Guangzhou, China) was used according to the instruction manual. The images were obtained on Nikon A1Si Laser Scanning Confocal Microscope (Nikon Instruments Inc, Japan).

**Measurement of ROS level**. The Beyotime Biotech (China) ROS assay kit was used to determine the levels of intracellular ROS. After treatment, chondrocytes were harvested and washed twice with PBS. Then the chondrocytes were centrifuged, and the supernatant was discarded. Finally, the chondrocytes were incubated with DCFH-DA (10 mmol/L) at 37 °C for 30 min in a darkroom for analysis by flow cytometry.

**Cell viability assay**. The cell viability was determined using the Cell Counting Kit-8 (CCK-8, Dojindo, Kumamoto, Japan). Chondrocytes were seeded into 96-well

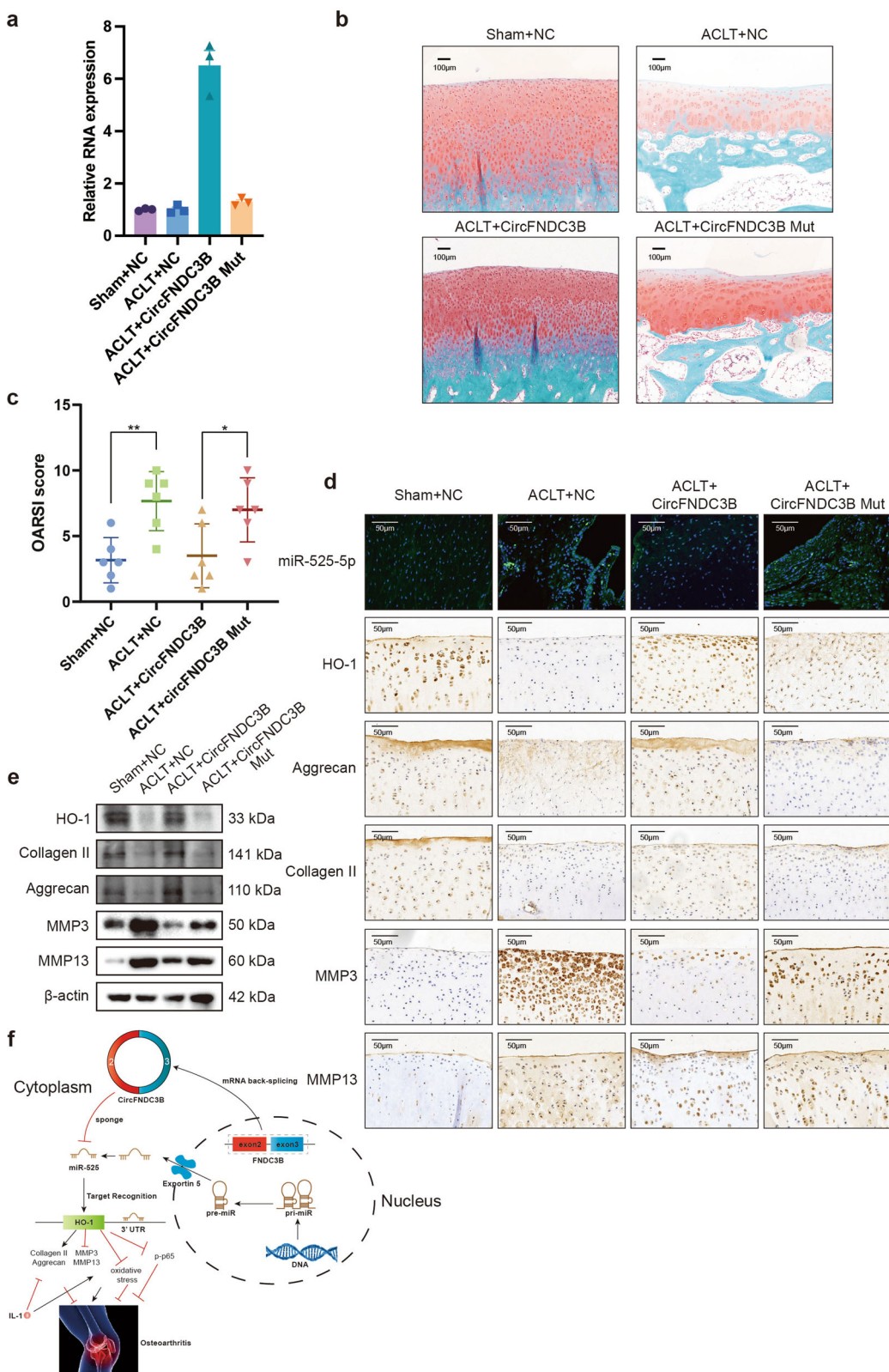

**Fig. 9 Injection of CircFNDC3B alleviates OA in a rabbit ACLT model. a**. RT-PCR analysis showed that the overexpression of CircFNDC3B in rabbit cartilage. ($n = 6$). **b**. Safranin-O/fast green staining of cartilage from 4 groups of rabbits. Scale bar, 100 μm. **c**. OARSI score used for the assessment of histological changes of rabbit knee cartilage. ($n = 6$). **d**. Histological analysis of rabbit knee cartilage by FISH and immunohistochemistry. MiR-525-5p, HO-1, MMP3, MMP13, Collagen II, and Aggrecan expression were examined. Scale bars, 50 μm, 100 μm. **e**. Western blot analysis of HO-1, MMP3, MMP13, Collagen II, and Aggrecan in each group. **f**. Schematic illustration of the circFNDC3B/miR-525-5p/HO-1 axis. Mean ± SEM; *$p < 0.05$; **$p < 0.01$; ***$p < 0.001$ by Student's t test.

plates at a density of $3 \times 10^3$ / well in triplicate. The CCK-8 was added to the wells at 24-, 48-, 72-, and 96-hours post-transfection. A microplate reader set at 450 nM (OD450) were used to measure the absorbance values of optical density (OD) in each well. Experiments were independently repeated three times.

**Statistical analysis**. The results were shown as mean ± standard error of the mean (SEM). Statistical analyses were performed using SPSS 22.0. Statistical significance ($P < 0.05$) was determined by ANOVA and Student's t-test, unless otherwise stated.

**Reporting summary**. Further information on research design is available in the Nature Portfolio Reporting Summary linked to this article.

## Data availability

RNAseq data are available in NCBI SRA database (SRA accession: PRJNA516555). Images of uncropped blots are provided in Supplementary Fig. S4. Source data for graphs are available in Supplementary Data 1. The other data that support the findings of this study are available from the corresponding author, upon reasonable request.

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

## Acknowledgements

This work was supported by National Key R&D Program of China (No. 2020YFC1107104), Medical Healthy Scientific Technology project of Zhejiang Province (WKJ-ZJ-1906, 2020384891), National Nature Science Fund of China (81972089, 81871797, 82001462, 82171560), Natural Science Fund of Zhejiang Province (LQ20H060005), No benefits in any form have been or will be received from a commercial party related directly or indirectly to the subject of this manuscript.

## Author contributions

S.W.F., Y.M. and P.H.S. conceived of the study and carried out its design. Z.Z.C., Y.Z.H., and Y.C. performed the experiments. Z.Z.C., X.D.Y., J.J.Z., and G.X. conducted the statistical analyses. Z.Z.C. wrote the paper. S.Y.S. and Z.A.H. revised the paper. All authors read and approved the final manuscript.

## Competing interests

The authors declare no competing interests.
