## [Peer Review File · Communications Biology]

Reviewers' comments:

Reviewer #1 (Remarks to the Author):

The authors describe the role of circFNDC3B on protecting osteoarthritis. The potential role of circFNDC3B in protective role on osteoarthritis has not been characterized yet. In this regard, the novelty and significance of this study are high in OA research field. They showed circFNDC3B – miR-525-5p – HO-1 pathway was one of the novel signaling pathway of OA protection. The results were very interesting and straightforward, but there are a few points that can improve this paper.

⟨Major concerns⟩

1. In Fig4, the authors show that repression of circFNDC3B upregulates the expression of miR-525-5p, which is its target. Although they show that miRs can be trapped as sponge, the meaning of expression change and trapping are different. Is there any involvement in transcription, maturation or instability? If circFNDC3B alters the expression of miR-525-5p, please discuss the mechanism by which it alters the expression.

2. In all histological images, the authors do not explain the details of the histology of cartilage. Information about the layer of cartilage is important in the pathogenesis of OA. Please explain the pathological features in more detail. Moreover, in Fig 9, Sham+NC and ACLT+CircFNDC3B only show the cartilage layer, while ACLT+NC and ACLT+CircFNDC3B show the subchondral bone. In Fig.9a, is it the same magnification? You should also show the weakly expanded image.

3. In Fig 7ab, HO-1 was highly expressed in LTP, compared with MTP. I think the data is very significant in comparing LTP and MTP. Furthermore, I think it would be more convincing if there were data on how it compares in normal and OA knees. I assume that the data shown in Fig. 1a was analyzed specifically for circR by RNaseR treatment. Do you have any information on normal RNAseq or database?

4. In Fig 8, the data presented by the authors are insufficient to show the involvement of the NFκB pathway. The authors claim that the phosphorylation of p65 is elevated, but it is not well discriminated by the band presented. Please evaluate quantitatively. Furthermore, evaluation of total p65 expression itself is also necessary to demonstrate p65 phosphorylation. (Antibodies for P-65 are listed in Supplementary Table S2.)

⟨Minor concerns⟩

1. Student's t-test is used for analysing between two groups. In the case of three or more groups, differences among the groups are assessed using a one-way analysis of variance. For significant results, specific between-group differences are determined using posthoc analysis. You should check all the statistical analyses again.

2. In materials and methods, you should describe in detail the conditions of your experiment and the equipment you used.

3. In all figures, there are many unclear areas, (especially in histology) so please check the resolution.

4. In Fig4, miR-93-3p also upregulated by circFNDC3B knockdown. Please discuss about this.

5. Line 171, Figure 5a is RT-qPCR, not WB, and Figure 5c is WB, not RT-qPCR.

6. In Fig 6a, the authors state that they identified genes associated with OA using pubmed, but they do not explain the criteria. If there are any criteria, please explain them in the Methods section.

7. In Fig7c, you should add western blot of HO-1.
8. In Fig 8, there is no explanation for "vector", please add it to the legend.
9. In line 341-344, you write that 16 rabbits underwent the ACLT operation and the remaining 8 underwent sham operation. Is this correct?
10. In Fig 9 and Discussion section, the authors state that circFNDC3B was elevated by the administration of AAV. Please explain how this was shown. Also, explain in detail the conservation between species.
11. In Fig 9 and Methods section, please include the information about this viral vector in detail.
12. There is a mixture of "si-circFNDC3B" and "circFNDC3B-si". Please unify them.
13. Line 607, β -actin is not a negative control, but a loading control.
14. In supplementary Figure S1, please change to a clearer picture.

Reviewer #2 (Remarks to the Author):

The study aimed to elucidate the function of a patient-derived Circular RNA (circRNA), circFNDC3B, in OA progression and its relationship with the NF- κ B signaling pathway and oxidative stress. The functions of circFNDC3B in OA were investigated in vitro and in vivo. The authors presented data which showed that circFNDC3B 39 promotes chondrocyte proliferation and protects the extracellular matrix (ECM) from degradation. CircFNDC3B inhibited oxidative stress in OA by regulating the circFNDC3B/miR-525-5p/HO-1 axis and presumably the NF- κ B signaling pathway. The authors suggested that overexpression of circFNDC3B alleviated OA in a rabbit model.

Key message: Identification of a new circFNDC3B/miR-525-5p/HO-1 44 signaling pathway, which delays OA progression in an ACLT OA rabbit model.

Complex study, which is of interest in the OA research field. The in vitro part is comprehensive and exp. Design is mostly appropriate and reflects the experience of the group. The in vivo part however is preliminary and data presentation is poor.

General points

- Quality of most of the figures, in particular the images showing immunofluorescence, histology and immunohistochemistry is poor and lettering in general is very often too small to be readable.
- Present representative uncropped western blot images for HO-1, collagen II, aggrecan, ADAMTS-5, MMP3, MMP13 and β -actin and include them as suppl. files. Indicate MW.
- Which forms of the MMPs (pro-, active, both) are recognized by the antibodies?
- The protein form of collagens are always indicated by Latin numbers, meaning it must be collagen II.
- According to WB data, MMP13 is mostly unaffected in all experiments. Please discuss that and try to explain.
- In order to clearly state that the CircFNDC3B/miR-525-5p/HO-1 axis interacts with the NF κ B pathway, the main component p50 should be included and phosphorylation status should be analysed.
- It is strange that the number of collagen II and aggrecan positive cells is that strongly affected by ACLT.
- To be correct, there need to be also SHAM groups included, which are treated with the CIRC FNDC3B and CIRC FNDC3Bmut virus. Of course, it is understandable for the sake of the 3R principle to spare animals. However, this is a clear limitation of the study and should be noted.
- Overall, the in vivo data are rather preliminary. Following data are missing in order to somehow complete the rabbit data:

- 1. Evidence for CIRC FNDC3B and CIRC FNDC3Bmut expression in the joint and by which tissue(s).
- 2. Analysis of the synovial fluid for these RNAs.
- Methods description often lacks details

Specific points

Abstract: What exactly means "patient-derived circRNA"? Was it isolated from patient tissue?

Introduction:

- OA is a whole joint, if not whole body disease and not a disease of the articular cartilage only! This should be made clear by rephrasing the first sentence and by adding one or more up to date references reflecting this dogma!

Results:

- Page 6, line 108: What exactly is meant with "new" human cartilage samples?
- Page 7, line 136: WB under fig.2C does not support the statement that the MMP13 band is stronger after circFNDC3B k.d. indicating increased protein synthesis. What nature is the second stronger MMP-13 band in the IL1 β lane?
- Page 8, lines 146-147: Again, MMP-13 is not affected by oe of circFNDC3B neither at the protein nor at the mRNA level.
- Page 9, line 171: figure 5a and 5c labelling is confused.
- Page 10, line 197: conclusion is wrong: Why should HO-1 have a role in OA pathogenesis if it is higher expressed in the LTP than in the MTP? Again, MMP13 bands are identical in WB under fig. 7c!
- Page 12: ADAMTS-5 expression is not affected. Please comment.

Methods

Human and rabbit cartilage and chondrocytes

- Ref. 20 is wrong and does not report about the isolation procedure of rabbit and human chondrocytes. Please provide details in your response.

Rabbit OA model

- Why were animals sacrificed after 7 weeks? Please provide rationale for that time point.

RNA extraction

- How many chondrocytes and in which passage were used for RNA extraction?
- How were chondrocytes cultured? In 2D or 3D?

WB

- How much protein was loaded?

Immunofluorescence

- Here, more details need to be provided, e.g. cell treatment, antibody dilutions etc. Ref. 22 does not describe immunofluorescence at all. Also, immunohistochemistry description is missing completely (fig. 9c).

- Was the same antibody dilution used for WB and IMF? If so, this is quite unusual.

- Antibodies need to be specified by including a catalogue number (table S2).

RNA overexpression.

- Here more details need to be provided. How was transfection efficiency? How was oe verified?

Figures

Fig. 6e: quality of WB image of HO-1 is poor. Should be replaced.

Fig. 7g: quality of WB image of aggrecan is poor. Should be replaced.

Fig. 8b,c: Please indicate clearly which forms of p65 are recognized? The phosphorylated and non-phosphorylated or both? It is common to show both forms in order to calculate the ratio of phosphorylated to non-phosphorylated form! Only then, activity changes can be calculated.

Fig.6

6a: Why are different magnifications shown for the 4 groups? Magnification should be comparable.

6c: miR-525-5p staining is not recognizable.

Reviewer #3 (Remarks to the Author):

In this article, the authors investigate the role of circFNDC3B from humans in articular chondrocytes and its relationship with the NF- κ B signaling pathway and oxidative stress. This study provides new insights into the response of articular chondrocytes to OA progression and further concretizes the role of circFNDC3B for alleviating OA in a rabbit model. This study is important to find a potential therapeutic target for the treatment of OA. However, some minor concerns are listed below.

- 1.Line 107-108: When the 10 cases and control cases were enrolled?
- 2.Line 144-146: Figure 3c appears to have errors. IL-1 β inhibits the expression of aggrecan, but the WB results are not consistent with the conclusion well.
- 3.Line 166: How did the authors measure luciferase activity?
- 4.Line 171: WB and RT-qPCR results are partly reversed
- 5.Line202-205: Figure 7g needs to be provided again. The Aggrecan bands are too blurred and the trend is not clear.
- 6.Line230-242: In vivo, how many were determined by WB, staining, and immunohistochemistry, respectively?
- 7.Line 341-342: For 24 rabbits involved, 16 rabbits were treated with ACLT and 8 rabbits were controlled as control.
Line 343-347, the 24 rabbits were again divided into 4 groups with every group containing 6 rabbits. These two places seem to be incompatible, please describe them in detail.

To Reviewer #1:

<major concerns>

1. In Fig4, the authors show that repression of circFNDC3B upregulates the expression of miR-525-5p, which is its target. Although they show that miRs can be trapped as sponge, the meaning of expression change and trapping are different. Is there any involvement in transcription, maturation or instability? If circFNDC3B alters the expression of miR-525-5p, please discuss the mechanism by which it alters the expression.

Authors: Apologies. Cytoplasm-localised circRNAs participate in translational regulation by acting as ceRNAs or coding RNAs. We performed AGO2 RIP assay, and the results revealed that circFNDC3B binds to AGO2 (figure 4A). This results showed that circFNDC3B functions by acting as ceRNAs. Bioinformatics analysis of circFNDC3B further revealed that it has no intrinsic ribosomal entry site (IRES) elements, circFNDC3B was not found to encode a protein (figure is as follows).

CircBank: IRES elements

ID	start	end	score
No matching records found			

Figure. The prediction of IRES elements of circFNDC3B.

2. In all histological images, the authors do not explain the details of the histology of cartilage. Information about the layer of cartilage is important in the pathogenesis of OA. Please explain the pathological features in more detail. Moreover, in Fig 9, Sham+NC and ACLT+CircFNDC3B only show the cartilage layer, while ACLT+NC and ACLT+CircFNDC3B Mut show the subchondral bone. In Fig. 9a, is it the same magnification? You should also show the weakly expanded image.

Authors: Thank you for the good point. Detailed explanations of the pathological features have been added in the manuscript. As for Fig 9b, the image is in same magnification. To avoid ambiguity, a new version of the image has been added to replace Fig. 9b.

3. In Fig 7ab, HO-1 was highly expressed in LTP, compared with MTP. I think the

data is very significant in comparing LTP and MTP. Furthermore, I think it would be more convincing if there were data on how it compares in normal and OA knees. I assume that the data shown in Fig. 1a was analyzed specifically for circR by RNaseR treatment. Do you have any information on normal RNAseq or database?

Authors: Thank you for the good point. We used GEO Profiles (NCBI) to find out the expression level of HO-1 in normal and OA knees. Profile GDS3758 / 203665_at showed that the expression level of HO-1 was higher in normal human chondrocytes than in osteoarthritis human chondrocytes, and the trends were similar between the matrix culture environment and monolayer culture environment (<https://www.ncbi.nlm.nih.gov/geoprofiles/66989213>).

4. In Fig 8, the data presented by the authors are insufficient to show the involvement of the NFkB pathway. The authors claim that the phosphorylation of p65 is elevated, but it is not well discriminated by the band presented. Please evaluate quantitatively. Furthermore, evaluation of total p65 expression itself is also necessary to demonstrate p65 phosphorylation. (Antibodies for P-65 are listed in Supplementary Table S2.)

Authors: Thank you for the good point. The images were added in Fig.8.

<Minor concerns>

1. Student' t t-test is used for analysing between two groups. In the case of three or more groups, differences among the groups are assessed using a one-way analysis of variance. For significant results, specific between-group differences are determined using posthoc analysis. You should check all the statistical analyses again.

Authors: Thank you for the good point. All the statistical analyses have been checked.

2. In materials and methods, you should describe in detail the conditions of your experiment and the equipment you used.

Authors: Thank you for the good point. We have added detailed descriptions of the experiment conditions and the equipment.

3. In all figures, there are many unclear areas, (especially in histology) so please check the resolution.

Authors: In accordance with your suggestion, the resolution was checked. The figures in the PDF file were compressed so it's not clear. The independent figure files are relatively clear.

4. In Fig4, miR-93-3p also upregulated by circFNDC3B knockdown. Please discuss about this.

Authors: Thank you for the good point. CircRNAs can sponge multiple miRNAs. In our research, we found that circFNDC3B can influence both miR-525-5p and miR-93-3p. WB was used to find out the influence of overexpression of these two miRNAs.

According to the results, we found that miR-525-5p can influence COL2A1, aggrecan, mmp3, mmp13 and ADAMTS-5 more pronounced than miR-93-3p (fig S2A). So we choose miR-525-5p rather than miR-93-3p.

5. Line 171, Figure 5a is RT-qPCR, not WB, and Figure 5c is WB, not RT-qPCR.

Authors: Apologies. The mistake has been corrected.

6. In Fig 6a, the authors state that they identified genes associated with OA using pubmed, but they do not explain the criteria. If there are any criteria, please explain them in the Methods section.

Authors: Apologies. The explanation has been added.

7. In Fig7c, you should add western blot of HO-1.

Authors: Apologies. The figure has been added.

8. In Fig 8, there is no explanation for "vector", please add it to the legend.

Authors: Apologies. The explanation has been added.

9. In line 341-344, you write that 16 rabbits underwent the ACLT operation and the remaining 8 underwent sham operation. Is this correct?

Authors: Apologies. A total of 32 rabbits involved. 24 rabbits were treated with ACLT and 8 rabbits were treated with sham surgery.

10. In Fig 9 and Discussion section, the authors state that circFNDC3B was elevated by the administration of AAV. Please explain how this was shown. Also, explain in detail the conservation between species.

Authors: Apologies. The explanation and qPCR results have been added.

11. In Fig 9 and Methods section, please include the information about this viral vector in detail.

Authors: Apologies. The information has been added in Fig S3. The sequence of circRNA was inserted into XbaI, which is a restriction enzyme cutting site.

12. There is a mixture of "si-circFNDC3B" and "circFNDC3B-si". Please unify them.

Authors: Apologies. The name has been unified.

13. In Line 607, β -actin is not a negative control, but a loading control.

Authors: Apologies. The mistake has been corrected.

14. In supplementary Figure S1, please change to a clearer picture.

Authors: Apologies. The pictures were taken during the surgery. Among all the pictures, these two pictures are relatively clear. We are sorry that our photography skills are not good enough. We added guide lines to the pictures. Hope this can make it easier to be noticed.

To Reviewer #2:

<General points>

1. **Quality of most of the figures, in particular the images showing immunofluorescence, histology and immunohistochemistry is poor and lettering in general is very often too small to be readable.**

Authors: Apologies. We found that the figures in the PDF file were compressed so it's not clear. The independent figure files are relatively clear. The font size has been enlarged.

2. **Present representative uncropped western blot images for HO-1, collagen II, aggrecan, ADAMTS-5, MMP3, MMP13 and β -actin and include them as suppl. files. Indicate MW.**

Authors: Thank you for the good point. The supplement files have been added.

3. **Which forms of the MMPs (pro-, active, both) are recognized by the antibodies?**

Authors: According to the instruction manuals of the antibodies we used, the antibodies recognized the C-terminal of the MMPs. So theoretically the antibodies can recognize both forms of MMPs.

4. **According to WB data, MMP13 is mostly unaffected in all experiments. Please discuss that and try to explain.**

Authors: Apologies. We did not notice this in our previous experiments. We purchased new batches of antibodies from Abcam and repeated the WB experiments. The new data showed that MMP13 was affected in experiments. Thus the old images have been replaced.

5. **In order to clearly state that the CIRC/NDC3B/miR-525-5p/HO-1 axis interacts with the NF κ B pathway, the main component p50 should be included and phosphorylation status should be analysed.**

Authors: Thank you for the good point. The images of these two have been added.

6. **It is strange that the number of collagen II and aggrecan positive cells is that strongly affected by ACLT.**

Authors: Apologies. We chose the most obvious images and put them in the figure. The old images have been replaced.

7. **To be correct, there need to be also SHAM groups included, which are treated with the CIRC/NDC3B and CIRC/NDC3Bmut virus. Of course, it is understandable for the sake of the 3R principle to spare animals. However, this is a clear limitation of the study and should be noted.**

Authors: Apologies. The limitation has been added in discussion section.

8. **Overall, the in vivo data are rather preliminary. Following data are missing**

in order to somehow complete the rabbit data: 1. Evidence for CIRC FNDC3B and CIRC FNDC3Bmut expression in the joint and by which tissue(s). 2. Analysis of the synovial fluid for these RNAs.

Authors: Apologies for our poor consideration, we forgot to put the results in the figure. The RT-qPCR results of circFNDC3B in rabbit cartilage has been added to Fig 9a. But we are so sorry that we did not collect synovial fluid when we harvested the joints from rabbits. We will add this limitation in discussion section.

<Specific points>

1. Abstract: What exactly means “patient-derived circRNA”? Was it isolated from patient tissue?

Authors: Sorry for the unclearness. The RNA sequence was based on patient-derived tissues, so we make such expression. The sentence have been corrected.

2. Introduction:

- OA is a whole joint, if not whole body disease and not a disease of the articular cartilage only! This should be made clear by rephrasing the first sentence and by adding one or more up to date references reflecting this dogma!

Authors: Apologies. The sentence has been rewritten.

3. Results:

- Page 6, line 108: What exactly is meant with “new” human cartilage samples?

Authors: Sorry for the unclearness. The “new” samples were meant to be different from the samples used for RNA sequencing. The sentence have been corrected.

- Page 7, line 136: WB under fig.2C does not support the statement that the MMP13 band is stronger after circFNDC3B k.d. indicating increased protein synthesis. What nature is the second stronger MMP-13 band in the IL1 β lane?

Authors: Apologies. We purchased new batches of antibodies from Abcam and repeated the WB experiments for three times. The new figure showed MMP13 was affected in experiments. Thus the old images have been replaced.

- Page 8, lines146-147: Again, MMP-13 is not affected by oe of circFNDC3B neither at the protein nor at the mRNA level.

Authors: Apologies. We did not notice this in our previous experiments. We purchased new batches of antibodies from Abcam and repeated the WB experiments. The new data showed that MMP13 was affected in experiments. Thus the old images have been replaced.

- Page 9, lane 171: figure 5a and 5c labelling is confused.

Authors: Apologies. The mistake has been corrected.

- Page 10: lane 197: conclusion is wrong: Why should HO-1 have a role in OA pathogenesis if it is higher expressed in the LTP than in the MTP? Again, MMP13 bands are identical in WB under fig. 7c!

Authors: Sorry about that. The mistake has been corrected.

- Page 12: ADAMTS-5 expression is not affected. Please comment.

Authors: Apologies. We purchased new batches of antibodies from Abcam and repeated the WB experiments for three times. The new data showed that ADAMTS-5 was affected in experiments. Thus the old images have been replaced.

4. Methods

Human and rabbit cartilage and chondrocytes

- Ref. 20 is wrong and does not report about the isolation procedure of rabbit and human chondrocytes. Please provide details in your response.

Authors: Apologies. We cited the wrong literature. The citation has been updated. Cartilage tissues were isolated from humans and rabbits, and treated with 0.2% type II collagenase (Sigma-Aldrich, USA) for 24 h at 37 ° C. After filtering through a 0.075 mm cell strainer and centrifugation at 1500 rpm for 10 min, the precipitated cells were cultured in Dulbecco's Modified Eagle Medium (DMEM) supplemented with 10% FBS (Thermo Fisher Scientific, Waltham, MA, USA). The culture was maintained in an incubator set to 37° C with 5% CO₂ and 100% humidity. The DMEM were changed and cells were washed with sterile phosphate buffered saline (PBS) every day during the first three days of culture. When the first-passage cells grown to occupy more than 80% of the bottom area of the culture dish, cells from one culture dish were digested with 1ml Trypsin and seeded into two culture dishes evenly as second-passage cells. The second-passage cells were used for transfection and other operations.

Rabbit OA model

- Why were animals sacrificed after 7 weeks? Please provide rationale for that time point.

Authors: According to the literature we found (PMID: 33580730), the researchers found that 8 weeks post-ACLT can induce significant progressive subchondral bone loss and proteoglycan loss. We therefore decided to set the required experimental period to 8 weeks. We did the surgery at week 0, did the injection at week 1, and sacrificed animals at week 8. The first week was to give time for the animals to recover, the total 8 weeks were to induce arthritis.

RNA extraction

- How many chondrocytes and in which passage were used for RNA extraction?

Authors: The cells used for RNA extraction were seeded into 6-well plates at a density of 2×10^5 cells/well. When the cells grown to occupy more than 80% of the bottom area of the 6-well plates, cells were given treatment. After the treatment, cells were ready for RNA extraction. Total RNA were extracted from

cartilage tissues using Ultrapure RNA Kit (CW BIO) (<https://www.cwbio.com/goods/index/id/10211>). The concentration and purity of the extracted RNA were measured using a Nanodrop-2000 spectrophotometer (260 and 260/280 nm) (Thermo Fisher Scientific, Waltham, MA).

- How were chondrocytes cultured? In 2D or 3D?

Authors: We cultured chondrocytes in 2D. Chondrocytes were cultured in culture dishes.

WB

- How much protein was loaded?

Authors: The protein samples were qualified the concentration of total proteins using Bicinchoninic acid (BCA) analysis (Beyotime, China). Then samples were diluted to 2 $\mu\text{g}/\mu\text{L}$. After adding 5* loading buffer and being heated in 100°C for 5-10 min, the samples were ready for WB. In each lane of the gels, 30 μg protein was added.

Immunofluorescence

- Here, more details need to be provided, e.g. cell treatment, antibody dilutions etc. Ref. 22 does not describe immunofluorescence at all. Also, immunohistochemistry description is missing completely (fig. 9c).

Authors: Detailed method has been added in manuscript.

- Was the same antibody dilution used for WB and IMF? If so, this is quite unusual.

Authors: The dilution used for IMF has been added in Table S2.

- Antibodies need to be specified by including a catalogue number (table S2).

Authors: The catalogue numbers has been added in Table S2.

RNA overexpression.

- Here more details need to be provided. How was transfection efficiency? How was oe verified?

Authors: The transfection efficiency and verification is in Fig 3A and B.

5. Figures

Fig. 6e: quality of WB image of HO-1 is poor. Should be replaced.

Authors: Apologies. The image has been replaced.

Fig. 7g: quality of WB image of aggrecan is poor. Should be replaced.

Authors: Apologies. The image has been replaced.

Fig. 8b,c: Please indicate clearly which forms of p65 are recognized? The phosphorylated and non-phosphorylated or both? It is common to show both forms

in order to calculate the ratio of phosphorylated to non-phosphorylated form!
Only then, activity changes can be calculated.

Authors: Apologies. The image of p65 and p-p65 has been added.

Fig. 6

6a: Why are different magnifications shown for the 4 groups? Magnification should be comparable.

Authors: Apologies. The image has been replaced.

6c: miR-525-5p staining is not recognizable.

Authors: Apologies. The image has been replaced.

To Reviewer #3:

1.Line 107-108: When the 10 cases and control cases were enrolled?

Authors: The cases were enrolled within three months of the start of the study.

2.Line 144-146: Figure 3c appears to have errors. IL-1 β inhibits the expression of aggrecan, but the WB results are not consistent with the conclusion well.

Authors: Apologies. We did not notice this in our previous experiments. We purchased new batches of antibodies from Abcam and repeated the WB experiments. The new data showed that aggrecan was affected in experiments. Thus the old images have been replaced.

3.Line 166: How did the authors measure luciferase activity?

Authors: For luciferase reporter analysis, HEK-293T cells were seeded into 48-well plates and cultured to 50%-70% confluence. A miRNA mimic or mimic-NC (RiboBio, Guangzhou, China) was co-transfected with the specific luciferase reporter plasmid into HEK-293T using Lipofectamine 3000 transfection reagent (ThermoFisher, USA) according to the manufacturer's instructions. Forty-eight hours after incubation, the luciferase activity was measured using a dual-luciferase reporter assay system (Promega, Madison, WI). The relative luciferase activity was determined using the value of hRLuc standardized to hFLuc.

4.Line 171: WB and RT-qPCR results are partly reversed

Authors: Sorry about that. The mistake has been corrected.

5.Line202-205: Figure 7g needs to be provided again. The Aggrecan bands are too blurred and the trend is not clear.

Authors: Apologies. The image has been replaced.

6.Line230-242: In vivo, how many were determined by WB, staining, and immunohistochemistry, respectively?

Authors: Apologies. In each ACLT rabbit, the bilateral knee joint was treated with ACLT surgery. Thus, for each group, there were 16 knee joints. They were divided evenly into different experiments.

7.Line 341-342: For 24 rabbits involved, 16 rabbits were treated with ACLT and 8 rabbits were controlled as control.Line 343-347, the 24 rabbits were again divided into 4 groups with every group containing 6 rabbits.These two places seem to be incompatible, please describe them in detail.

Authors: Apologies for the unclearness. A total of 32 rabbits involved. 24 rabbits were treated with ACLT and 8 rabbits were treated with sham surgery.

Reviewers' comments:

Reviewer #1 (Remarks to the Author):

Basically, the authors answer all the reviewer's questions.

However, I have the following two concerns.

1. The statistics are not well described. It does not state which statistical method was used in which analysis. Please describe them in the figure legend or something.

2. The circR plasmid design was available. It seems that the circR plasmid is just a vector of AAV with the mature circR sequence. The authors have not confirmed that the product produced by this virus is circular.

Therefore, I think the authors should check using RNaseR treatment. If there is a sponge function even in the linear state, it is necessary to mention it.

Reviewer #2 (Remarks to the Author):

The authors responded to most of my comments fine. A few comments would need additional attention.

2) Western blot images from collagen II and Ho-1 WB are NOT uncropped! Please add appropriate WB images to suppl.files.

3) Please add the information about the MMP recognition site to the appropriate method part.

4. Methods

Rabbit OA model: please include information about timeline into the text.

RNA extraction: Please include the information into the text.

Chondrocyte culture: Please include info into text.

WB: Please include info into text

To Reviewer #1:

1. The statistics are not well described. It does not state which statistical method was used in which analysis. Please describe them in the figure legend or something.

Authors: Apologies. The statistical method has been added into the figure legend.

2. The circR plasmid design was available. It seems that the circR plasmid is just a vector of AAV with the mature circR sequence. The authors have not confirmed that the product produced by this virus is circular.

Therefore, I think the authors should check using RNaseR treatment. If there is a sponge function even in the linear state, it is necessary to mention it.

Authors: Thank you for the good point. The RNase R experiment has been added in Supplementary Figure S2.

To Reviewer #2:

1. Western blot images from collagen II and Ho-1 WB are NOT uncropped! Please add appropriate WB images to suppl. files.

Authors: Apologies. The uncropped western blot images have been added to supplementary files.

2. Please add the information about the MMP recognition site to the appropriate method part.

Authors: Apologies. The information about the MMP recognition site has been added to method. According to the information page from Abcam (<https://www.abcam.cn/mmp13-antibody-ep1263y-ab51072.html> and), the MMP13 recognition site is synthetic peptide within Human MMP13 (C terminal), and the MMP3 recognition site is synthetic peptide within Human MMP3 aa 450 to the C-terminus (C terminal). Abcam claims that the exact sequence is proprietary.

3. Methods:

Rabbit OA model: please include information about timeline into the text.

Authors: Apologies. The timeline picture has been added in Supplementary Figure S1c.

RNA extraction: Please include the information into the text.

Authors: Apologies. We have rewritten a new detailed version of this experiment.

Chondrocyte culture: Please include info into text.

Authors: Apologies. The detailed description has been added into the method.

WB: Please include info into text

Authors: Apologies. A detailed version has been added into the method.

To Reviewer #3:

1.Line 107-108: When the 10 cases and control cases were enrolled?

Authors: The cases were enrolled within three months of the start of the study.

2.Line 144-146: Figure 3c appears to have errors. IL-1 β inhibits the expression of aggrecan, but the WB results are not consistent with the conclusion well.

Authors: Apologies. We did not notice this in our previous experiments. We purchased new batches of antibodies from Abcam and repeated the WB experiments. The new data showed that aggrecan was affected in experiments. Thus the old images have been replaced.

3.Line 166: How did the authors measure luciferase activity?

Authors: For luciferase reporter analysis, HEK-293T cells were seeded into 48-well plates and cultured to 50%-70% confluence. A miRNA mimic or mimic-NC (RiboBio, Guangzhou, China) was co-transfected with the specific luciferase reporter plasmid into HEK-293T using Lipofectamine 3000 transfection reagent (ThermoFisher, USA) according to the manufacturer's instructions. Forty-eight hours after incubation, the luciferase activity was measured using a dual-luciferase reporter assay system (Promega, Madison, WI). The relative luciferase activity was determined using the value of hRLuc standardized to hFLuc.

4.Line 171: WB and RT-qPCR results are partly reversed

Authors: Sorry about that. The mistake has been corrected.

5.Line202-205: Figure 7g needs to be provided again. The Aggrecan bands are too blurred and the trend is not clear.

Authors: Apologies. The image has been replaced.

6.Line230-242: In vivo, how many were determined by WB, staining, and immunohistochemistry, respectively?

Authors: Apologies. In each ACLT rabbit, the bilateral knee joint was treated with ACLT surgery. Thus, for each group, there were 16 knee joints. They were divided evenly into different experiments.

7.Line 341-342: For 24 rabbits involved, 16 rabbits were treated with ACLT and 8 rabbits were controlled as control.Line 343-347, the 24 rabbits were again divided into 4 groups with every group containing 6 rabbits.These two places seem to be incompatible, please describe them in detail.

Authors: Apologies for the unclearness. A total of 32 rabbits involved. 24 rabbits were treated with ACLT and 8 rabbits were treated with sham surgery.

REVIEWERS' COMMENTS:

Reviewer #1 (Remarks to the Author):

I appreciate the through revision the author made. The authors revised the manuscript according to the comments by all reviewers. The quality and impact of the presented data are sufficient to merit publication of the manuscript in Communications Biology.

Reviewer #2 (Remarks to the Author):

The authors have addressed all remaining comments fine.

There is only a minor comment left: COI2A1 and ACAN are the gene names and should be replaced by the protein names throughout the manuscript including the figures: Collagen II and Aggrecan

To Reviewer #1:

I appreciate the through revision the author made. The authors revised the manuscript according to the comments by all reviewers. The quality and impact of the presented data are sufficient to merit publication of the manuscript in *Communications Biology*.

Authors: We would like to express our great appreciation for your patience and efforts.

To Reviewer #2:

1. The authors have addressed all remaining comments fine.

Authors: Thank you for your good points and great efforts.

2. There is only a minor comment left: COL2A1 and ACAN are the gene names and should be replaced by the protein names throughout the manuscript including the figures: Collagen II and Aggrecan

Authors: Apologies. The names have been replaced.